# Towards clinical applicability of fMRI via systematic filtering

**Jan Willem Koten[1,2]°\*, André Schüppen[2]°, Guilherme Wood[1], Martin Holler[3]**

**1** Institute of Psychology, University of Graz, Graz, Austria, **2** IZKF - Interdisciplinary Center for Clinical Research, RWTH Aachen University, Aachen, Germany, **3** Department of Mathematics and Scientific Computing, University of Graz, Graz, Austria

☯ These authors contributed equally to this work.
\* jan.koten@uni-graz.at

## Abstract

It is a common practice to evaluate the reproducibility of fMRI at the group level. However, for clinical applications of fMRI, where the focus is on reproducibility of single individuals, the high test-retest reliability that is sometimes reported for group-based measures can be misleading. On the level of single subjects, reproducibility of fMRI is still far too low for clinical applications, not even meeting the standards to use fMRI for scientific purposes. The goal of this work is to enhance the poor single-subject time course reproducibility of fMRI. For this purpose, we have developed a framework for post-processing fMRI signals using Savitzky-Golay (SG) filters in conjunction with general linear model (GLM) based data cleaning. The parameters of these filters were trained to be the optimal ones based on a dataset of working memory relevant signals. By employing our data-driven filtering framework, we successfully improve the average reproducibility correlation of a single fMRI time course from r = 0.26 (as obtained with a conventional statistical parametric mapping (SPM) data cleaning pipeline) to a fair level of r = 0.41. Additionally, we are able to enhance the average connectivity correlation from r = 0.44 to r = 0.54. Our conclusion is that signal post-processing with a data-driven SG filter framework may substantially improve time course reproducibility compared to conventional denoising pipelines. As a conservative estimate, we conjecture that roughly 10–30% of the population may benefit from optimized fMRI pipelines in a clinical setting depending on the measure of interest while this number was nihil for conventional fMRI pipelines.

## Introduction

Test-retest reliability guidelines, which are based on the intra-class correlation (ICC), have been established to aid in the interpretation of fMRI reproducibility in a clinical context. An ICC value below 0.40 is considered poor, 0.40–0.59 is fair, 0.60–0.74 is good, and above 0.75 is excellent [1–3]. For scientific purposes, a fair test-retest

**Data availability statement:** The minimal dataset that includes only the fMRI time courses used for the analysis reported in this paper is provided in the supporting information. The entire study can be replicated with the minimal data set provided. The minimal data can be analyzed with the Matlab code CleanBrain https://github.com/hinata2305/CleanBrain.

**Funding:** This study was supported by the, "Fonds zur Förderung der wissenschaftlichen Forschung" (grant number: P 22577-B18). The authors acknowledge the financial support by the University of Graz.

**Competing interests:** The authors have declared that no competing interests exist

reliability of at least 0.4 is suggested, while an excellent correlation of at least 0.75 is required for clinical purposes. A value of at least 0.6 is considered good reproducibility, which may allow fMRI to be used as an "ancillary diagnostic tool" in conjunction with existing clinical routines.

Recent meta-analysis studies focusing on the group reproducibility of task-based fMRI (ICC = .40) and resting state fMRI (ICC = 0.29) suggest that functional connectivity fMRI (fc-fMRI) may not be suited for clinical purposes [2,3]. In addition, if the reported group reproducibilities are truly informative for clinical applications of fMRI is debatable, as group reproducibility does not necessarily reflect single subject reproducibility. While scientists focus on group reproducibility, clinicians are concerned with the reproducibility on the level of single subjects.

Single-subject fMRI time course reproducibility is poorly investigated although it is critical for understanding brain activity and connectivity as it limits the height of beta values in brain activity studies and the strength of path correlations in connectivity studies. Single subject time course reproducibility is at best 0.25 [4,5] suggesting that the connectivity among regions cannot exceed this value [6,7]. This limits the usefulness of fMRI for diagnostic purposes. In clinical settings, methods like structural MRI, direct brain stimulation, or single-cell EEG recording are considered the gold standard. Diagnostic fMRI is often seen as an additional tool that is not part of routine clinical practice and may not be covered by insurance companies [8,9]. A recent meta-analysis nonetheless suggests that conducting fMRI mapping prior to a surgical procedure reduces the likelihood of functional deterioration afterward (odds ratio: 0.25; 95% CI: 0.12, 0.53; P < .001) [10]. In this work, we consider the question to what extend single subject test-test reliability in fMRI, and thus its clinical applicability, can be improved via optimized signal processing pipelines. A significant improvement of single subject test-retest reliability might potentially improve the odds ratio reported above.

A direct way to improve time course reproducibility is by reducing noise levels through filters. However, classic low-pass filters that are used to remove high frequency noise from the fMRI signal increase autocorrelations of time courses, which is problematic for event-related designs that analyze rapid cognitive changes in time courses [11]. Moreover, low pass filters may also induce spurious connectivity correlations [12]. For these reasons, the classic low-pass filters, specifically the HRF and Gaussian filters that were commonly used in SPM 99, were removed in subsequent versions of the package, including SPM 5, SPM 8, and SPM 12. Consequently, low-pass filters are currently rarely utilized in task-driven fMRI studies, although they are still applied to resting-state data. The question arises as to whether it is possible to develop low-pass filters that effectively remove noise while preserving the cognitive signal of interest and also maintain autocorrelations at acceptable levels.

Savitzky-Golay (SG) filters are commonly used to separate noise from signal [13]. The SG filters work by replacing the central point of a signal in a moving window of size n with the value of a polynomial function that best fits the observed signal within that window. They can uphold acceptable levels of autocorrelation, making them

promising for enhancing the reproducibility of single time courses [14]. However, SG filters comprise parameters (such as window size and order of the polynomial approximation) for which an optimal choice for fMRI purposes is not well understood, but whose values strongly influence filtering performance. Few studies have incorporated SG filters in fMRI signal processing, and the rationale behind the parameter choices are not documented [15,16]. Additionally, SG filters have not been established to remove high frequency noise. To address this issue, it has been suggested to optimize the parameters of SG filters using brute force algorithms [15].

In this study, our objective is to determine the optimal filter parameters. We aim to accomplish this by maximizing the correlation between the filtered time course (observed time course) and an empirically derived predictor time course, while also maintaining autocorrelations of the time course within a predefined limit.

To establish the empirically derived predictor time course, we calculate a subject-specific hemodynamic response function (HRF). This is done by averaging the task-related signal changes in the time course. This approach takes into account the individual variations in BOLD expression between subjects, resulting in a more accurate representation of brain activity as compared to using a canonical hemodynamic response function that is usually obtained through mathematical modeling. Furthermore, the predetermined limit for autocorrelations is derived from the empirical predictor time course. By leveraging the autocorrelations of the predictor time course, which are believed to approximate the true autocorrelation of the time course, we ensure that the filtered time course remains within acceptable boundaries. Finally, we assess if the filter constellations developed truly enhance the test-retest reliability of individual subject time courses, as well as the detectable connectivity in a distinct cognitive experiment. A key aspect of this evaluation will involve examining whether the residual noise time courses contain signals in task related frequency bands. This will help to determine whether our approach inadvertently removed any cognitive components from the fMRI signal.

## Methods

### Ethics statement

All subjects gave their written informed consent to participate in research before the actual experiment took place. All investigations were conducted according to the principles expressed in the Declaration of Helsinki. The study was approved by the Ethics committee of the University of Graz under GZ 39/31/63 ex 2011/12.

### Participants

The 67 individuals studied, varied in age and academic attainment and are more or less representative for the age and sex distribution of the population in Austria [17]. All individuals were free from psychiatric or neurological diseases and provided written informed consent. The recruitment of the subjects started at august 2012 and ended at February 2014.

### Working memory tasks

Prior to scanning, participants were trained in the tasks under study until they performed confidently. These tasks involved retaining letters (verbal Sternberg task) or spatial positions (spatial Sternberg task) in working memory while performing a number Stroop task. The verbal Sternberg task required identifying the identity of probe and target letters as identical or not, while the spatial Sternberg task required judging the position of probe and target as identical or not. The main task frequency observed during analysis was at 0.04 Hz, while the cognitively relevant shifts within the task occurred in a higher frequency band (~0.04Hz -0.25Hz). More details about the working memory task can be found in the S1 Text, while S1 Fig reports the relationship between the task and the empirical BOLD response for different pipelines. The latter figure shows that very large individual differences in the BOLD response exist which speaks against the use of the canonical HRF to construct predictor time courses, and motivates our choice to use an empirically derived predictor time course.

## Co-registration, Grey matter time-course extraction and nuisance regressors

Preprocessing of the various experiments was performed with the FreeSurfer/FsFast pipelines. In a first step, MRI scans of the individuals were segmented into gray matter and white matter, while the dura mater and pia mater were removed using the recon-all pipeline of the FreeSurfer package. The 3D gray matter time courses from the test and retest sessions were brought into 2D FS_average space using FsFast options, which utilize the spherical alignment methods of FreeSurfer. Subsequently, 34 regions of interest related to working memory, derived from a meta-analysis, were spherically aligned with the FS_average mesh space [18]. The denoising pipeline of FSFAST closely resembles the Comp-Cor pipeline [19]. FsFast extracts noise time courses from the white matter and ventricular systems, which have been accurately isolated by the FreeSurfer pipeline. The entire set of time courses of the respective tissues undergo principal component analysis for each individual subject when the fcseed-config -wm -fcname wm.dat -fsd bold -pca -cfg wm.config fcseed-sess -s sessionid -cfg wm.config OR fcseed-config -vcsf -fcname vcsf.dat -fsd bold -pca -cfg vcsf.config fcseed-sess -s sessionid -cfg vcsf.config commands of FsFast are used (https://surfer.nmr.mgh.harvard.edu/fswiki/FsFastFunctionalConnectivityWalkthrough). The top five principal components from each tissue type are extracted from the WM.dat (white matter) respective VCSF.dat (ventricles) files and subsequently used as regressors to denoise the data for every run and each subject within our own pipeline. Additionally, the FsFast pipeline subjects the 3 translations and 3 rotation motion parameters obtained during the motion correction procedure to principal component analysis (https://surfer.nmr.mgh.harvard.edu/fswiki/FsFastTutorialV6.0/FsFastPreProc). We selected the top two principal components (PC) from the orthogonalized FSFAST mcprextreg matrices and combined them with the five components from white matter and the five components from the ventricles, resulting in a total of 12 regressors for denoising purposes.

## Scanner parameters

MRI scans were performed on a 3T Siemens Magnetom Skyra (Siemens Medical Systems, Erlangen, Germany) equipped with a 32-channel head coil. We used a 3D-MPRAGE sequence (176 slices per slab, FOV = 256 mm, TR = 2530 ms, TE = 2.07 ms, TI = 900 ms, Flip angle = 9°, voxel size = 1 mm isotropic). Functional imaging data were obtained using a Siemens Grappa parallel acquisition scheme with pat factor 2; using following parameters Flip Angle 72 degrees, TR = 1240 ms, TE = 30 ms. Volume dimensions were 64*64*23, with voxel resolution 4*4*4 mm with a gap of 10% For the working state analysis in total 487 volumes were obtained per task per run.

## Nomenclature

In this project, multiple signal processing components are utilized. To ensure clarity, we will define each component. Firstly, *time course denoising* involves eliminating physiological sources of noise like heart rate and respiration artifacts and head motions from grey matter time courses. This is achieved by incorporating nuisance regressors such as white matter, ventricle, and head motion time courses into a General Linear Model (GLM). *Detrending*, on the other hand, focuses on removing the slow signal trends as captured by the SG and SPM filters from the grey matter time courses. When both denoising and detrending are simultaneously performed within a GLM, we will refer to the resulting time courses as "denoised and detrended". Finally, high-frequency noise is eliminated from the data using an SPM or SG filter. This can be done either by incorporating the filter into the GLM along with the 12 nuisance regressors and slow signal trend or after the completion of data denoising and detrending. We will refer to this process as "*cleaning*". We will construct an empirical *predictor time course* from hemodynamic responses of the single subjects and compare the results to the (un)filtered time courses that we refer to as *observed time courses*. Finally, we will subtract the SPM or SG cleaned time-courses from the paired denoised and detrended time-courses per subject. The resulting time courses that mainly contain high frequency noise are referred to as *residual noise time courses*.

## General overview of the experiment and MATLAB ® packages

The SG filters were optimized with a verbal working memory task and validated with a spatial working memory task. The method was developed on the basis of in-house MATLAB® software package called CleanBrain [20]. The results in this paper were obtained with MATLAB version 2021b. One script called "find_filter.m" was used to optimize the SG parameters for both the removal of signal drifts and high frequency noise; we will refer to this as the optimization experiment. The second script, called "test_filter.m" was used to evaluate the effects of the proposed pre-processing method on test-retest reliability, connectivity, detectable connectivity, autocorrelations of time-courses and Fourier spectra of noise time courses. We will refer to this as validation experiment. We tried to avoid circularity or self-referential reasoning on two levels. First, for the filter optimization procedure, a predictor time course was obtained from a test set and correlated with an SG filtered time course obtained from a retest set and vice versa. Second, the optimal filter parameters obtained from a verbal task were validated in a spatial task. For an overview of the overall pipeline, we refer to Fig 1.

## Methods for the optimization experiment.

### Background on SG filters

The Savitzky-Golay filter is a type of smoothing filter used to remove noise from a signal. It works by fitting a polynomial curve to a set of adjacent data points, which is then used to smooth out the signal. The filter can be described mathematically as follows:

$$y(l) = a0 + a1\ l + \ldots + an\ ln$$

where "y" is the filtered signal, "l" is the index of the data point, and "a0…an" are the coefficients of the polynomial curve. The coefficients can be calculated using the least-squares method, which involves minimizing the sum of the squared differences between the polynomial curve and the original data. The filter window size and the order of the polynomial curve can be adjusted to control the degree of smoothing.

The sliding window needs adjustments at the very beginning and the very end of time-courses. An accepted solution is to artificially extend data by adding reverse copies of the first and last data points at the appropriate places of the time-series [13]. We employed this procedure here and extended the data by adding, in reverse order, replicates of the first $(m-1)/2$ points at the beginning and replicates of the last $(m-1)/2$ points at the end, where $m$ denotes the size of the window in use.

### Filter optimization

The SG filters were engineered to remove slow signal drifts (detrending) and high-frequency noise (cleaning) using in-house MATLAB® scripts as follows: For removing slow signal drifts, the signal was filtered with an SG filter such that only low frequency trends remained. This low frequency signal, together with twelve nuisance regressor (five estimates of white matter noise, five estimates of ventricle noise and two estimates of head motion) was then used to denoise the grey matter time courses within a GLM framework. For removing high frequency noise, the SG filters were directly applied as noise filters on the previously filtered signal. The parameters of the filters were optimized as follows:

***Optimization for removing slow signal drifts (detrending)*** To optimize the parameters of the SG filters for removing slow signal drifts, a brute force search was conducted to find the optimal window size and polynomial degree. In this process, the empirical predictor time course was used as ground truth and correlated with the observed (filtered) time course per region for each subject. To obtain the predictor time courses per brain region, the grey matter time courses were denoised within a standard GLM framework. This framework included 5 white matter regressors, 5 ventricle regressors, and 2 motion regressors obtained through principal component analysis. Subsequently, the "event-related average" or empirical HRF was obtained by averaging the 24 sections of the denoised time course related to the cognitive events, including the resting baseline condition. It should be noted that the baseline was jittered, meaning that only the parts of the baseline shared among the 24 events were extracted. Finally, the predictor time course was constructed by concatenating

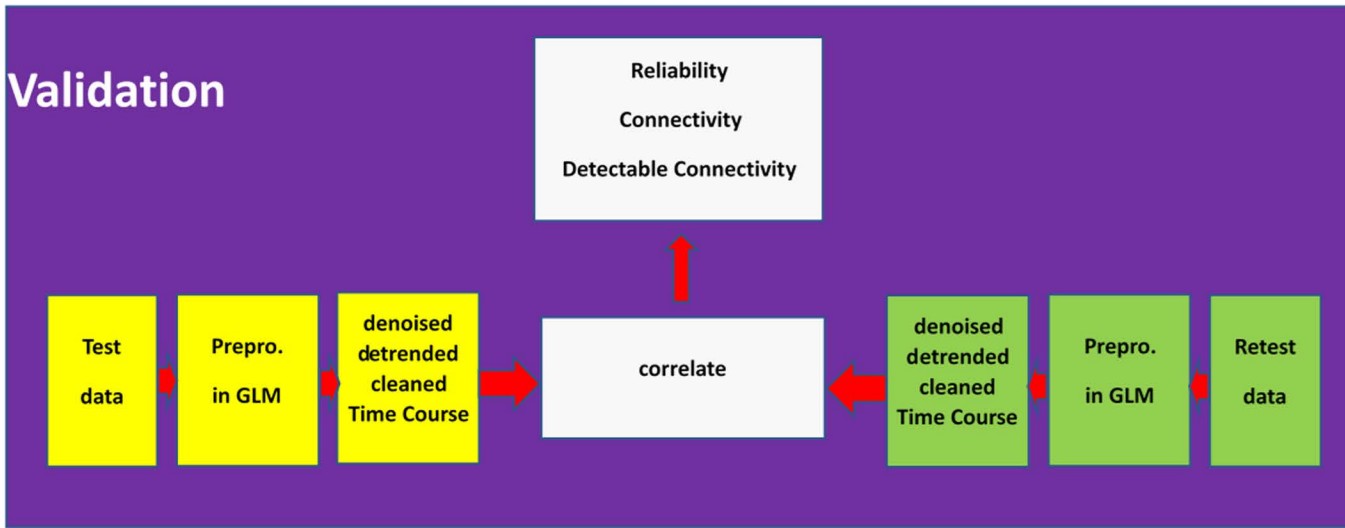

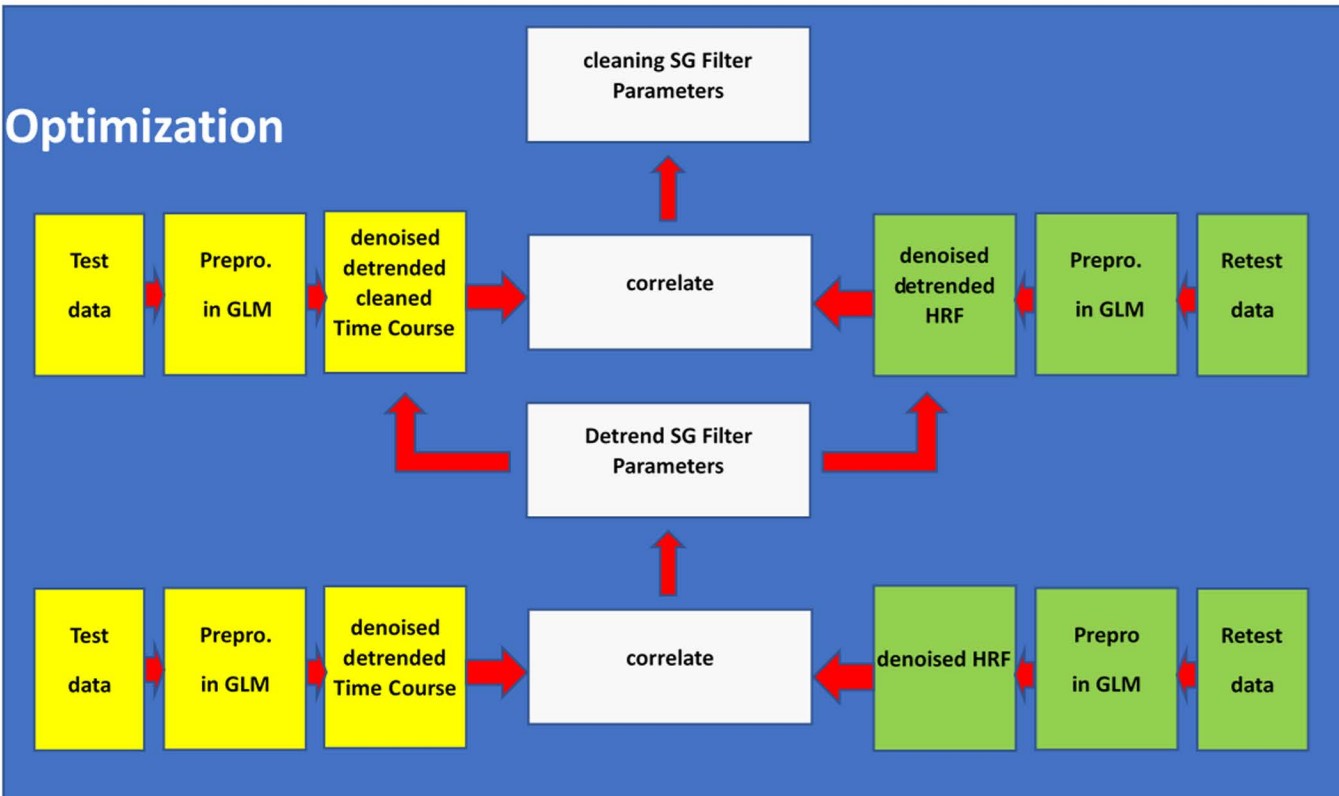

**Fig 1. Our pipeline consisted of two data sets: one for optimization purposes and one for validation purposes, each data set was measured twice (test and retest). The purpose of the optimization set was to determine the optimal parameters for the SG filters through a brute force approach. The validation set was used to assess the impact of the SG filters on connectivity and reliability. To optimize the SG detrend filters, we correlated the SG treated time courses (observed time course in yellow) with the HRF (predicted time course in green) obtained from the retest data and vice versa. The detrend filters were then used as input for the second step in which the clean filters were developed. Note that we only depicted HRF for the retest run for demonstration purposes. However as mentioned in the methods we also used the pipeline the other way around.**

the "event-related average" 24 times. The observed time courses were denoised and detrended using SG filters, which were incorporated within the previously described GLM framework. The search space for the filter parameters consisted of a window size ranging from 3 to 487 in steps of two, and the polynomial degree, for each window size, ranging from 1 to window size - 1. Finally, the tiny fractions of the baseline that were not shared among the events were omitted such that the observed timcourses could be correlated with the predictor time course. As the predictor time course of the test and retest runs were not identical, the predictor time course was estimated from the test run and correlated with the SG filtered time courses from the retest run, and vice versa. Each iteration resulted in 4556 correlations (2 runs * 34 time courses * 67 participants). The overall optimization pipeline is summarized in Fig 2, and the specific effects of the pipeline on the time courses of interest are visualized in S2 Fig.

*Optimization for time course cleaning* To optimize the parameters of the SG filter for the cleaning step, we utilized a brute force search approach to maximize the correlation between the predicted and observed time-courses. In order to obtain the predictor time course, we processed the data in the same manner as before, but additionally utilized the SG detrending filter that was obtained in the previous optimization step. The observed time-courses were processed in a similar fashion, with the inclusion of an additional SG cleaning filter that was applied after the denoising and detrending process had been completed within the GLM. The parameter search space for the SG filters remained the same as in the previous step, with the exception that polynomial orders larger than 50 were excluded from consideration. In the cleaning step, we not only focused on correlating the observed and predictor time courses for parameter optimization, but also took into account constraints on the auto-correlation structure of the observed signal after cleaning. To obtain a ground-truth autocorrelation structure, we computed the lag 1–4 autocorrelation of the predicted time courses that were detrended with an SG filter (window = 311; polynomial = 40) as this filter yielded optimal results. For the observed time courses, we averaged the lag one to four autocorrelations. Subsequently, we calculated the root mean square error (RMSE) between the four predicted and observed autocorrelations. To identify filters with appropriate autocorrelation behavior, we masked the correlations between the predicted and observed time-courses using an RMSE threshold of less than 0.1. Fig 3 provides a summary of the steps involved in optimizing the SG parameters for cleaning while the specific effects of the pipeline on the time course of interest is visualized in S3 Fig. We note that, as alternative to this second step, we also tried to optimize SG filters for cleaning directly within a GLM framework together with the nuisance and slow trends, but this resulted in poor results that are the therefor only reported in the S2 Text [21].

## Methods for the validation experiment

To evaluate the detrending and cleaning of fMRI time-courses with SG filters, we compared the following approaches.

*For detrending* three different filtering methods were implemented: a "short" Savitzky-Golay (SG) filter with a window of 69 TRs and a polynomial order of 6 (69/6), a "long" SG filter with a window of 311 TRs and a polynomial order of 40 (311/40) (these two parameter sets were selected on the basis of the optimization experiment) and a standard SPM detrending filter with a cutoff frequency of 0.0078 Hz (this serves as reference method). Slow signal trends were simultaneously incorporated with the nuisance components within the GLM framework as before.

*For cleaning* time-courses that underwent detrending with the SPM filter were subjected to either a hemodynamic-response-function (HRF)-based filter or a Gaussian smoothing kernel with a width of 2.48 sec [11]. We utilized the original SPM Matlab script spmfilter.m, which calls spmhrf.m, spmdctmtx.m, and spmGpdf.m, to implement the filtering. Time-courses detrended with a SG (69/6) filter were subjected to a SG (15/8) cleaning filter, whereas time-courses detrended with a SG (311/40) filter were subjected to a SG (3/1) cleaning filter which equals a simple moving average. We made the comparisons between the two SG cleaning filters because we wanted to examine whether polynomial filters (15/8) outperform simple moving averages (3/1). In summary following pipelines were considered (see also Fig 4 for additional information):

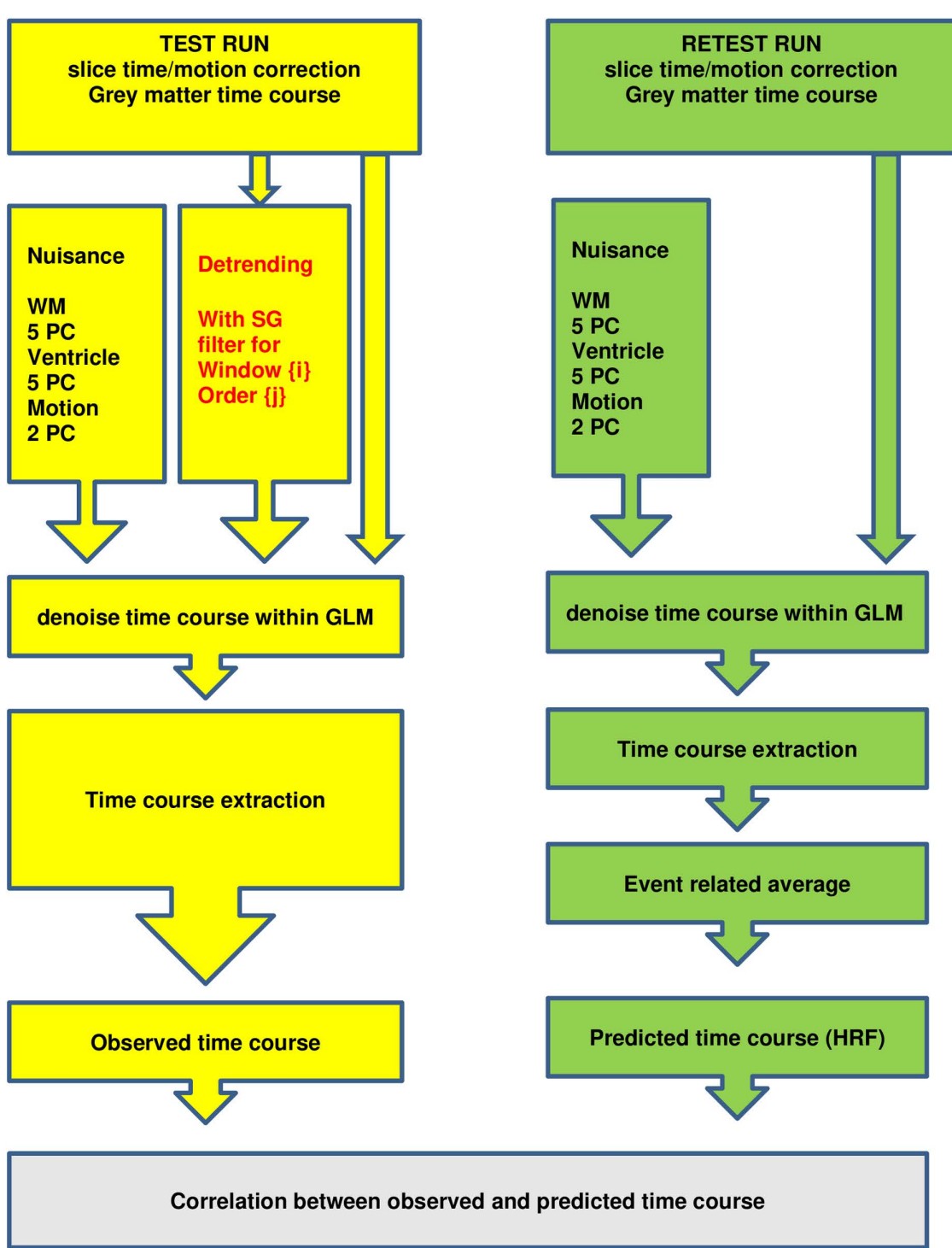

**Fig 2. Brute force pipeline which optimized SG filter parameters for detrending purposes. Red text refers to SG parameters that were optimized via this pipeline.** The parameter search space comprised a window size from 3 to 487 in steps of two, and the polynomial degree, for each window size, from 1 to window size -1. To optimize the parameters, the SG-processed observed time courses were correlated with the predicted time courses. Yellow and Green parts are in agreement with the pipeline elements depicted in Fig 1. Abbreviations: WM = white matter; PC = principal components; GLM = general linear model.

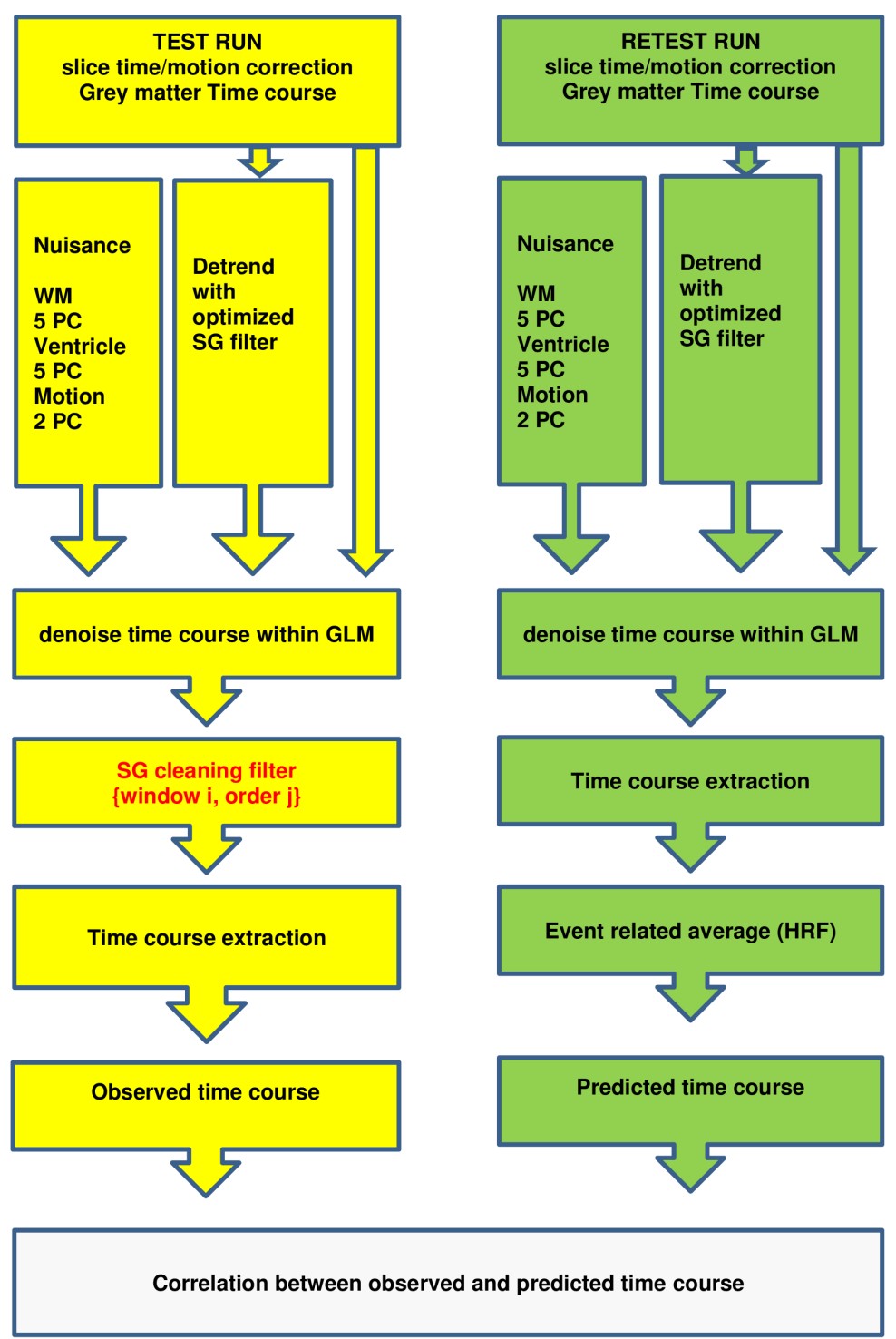

**Fig 3. Brute force pipeline for the second phase of the experiment which optimized SG parameters for cleaning.** Note that the optimal SG detrending parameters found in phase one were applied in phase 2. Red text refers to SG parameters that were optimized via this pipeline. The parameter search space for the SG filters was the same as in the previous step, with the restriction that polynomial orders larger than 50 were not considered. Yellow and green parts of the graph refer to the convention established in Fig 1.

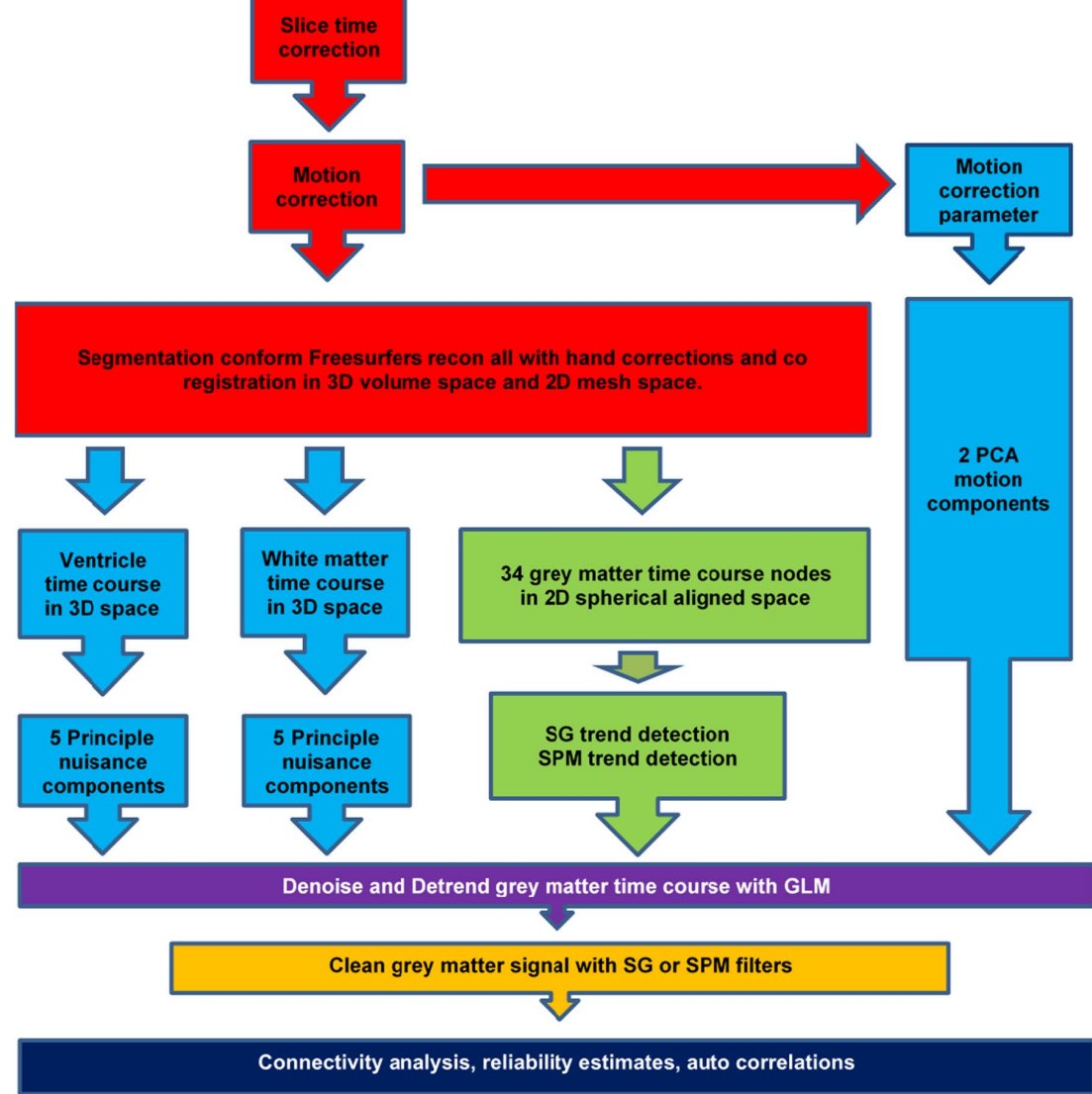

**Fig 4. A description of the validation pipeline.** Red refers to the "raw data". Light blue refers to denoise elements. Green refers to detrending. Yellow refers to cleaning filters. Purple refers to weighted noise and trend removal within a GLM frame work. While dark blue refers to the analysis module. The colors of the letters in the table refer to the colors of the pre-processing depicted in Figure. Abbreviations: SG = Savitzky-Golay. Results for the distinct pre-processing pipelines are given in Table 1-2.

## Simple Pipeline

***raw***: Slice time correction + motion correction

***denoise***: Denoising with 12 nuisance regressors in GLM (5 WM PC, 5 Ventricle PC, 2 motion PC)

## SPM Pipelines

***detrend SPM:*** Denoised and detrended with 13 nuisance regressors in GLM detrend filter = Discrete cosine transform filter set at 128 seconds (performed with spm_filter.m)

**clean HRF:** Denoised and detrended (DCT 128 sec) with 13 nuisance regressors in GLM + SPM HRF filter SPM HRF = Fourier filter based on HRF (performed with spm_filter.m)

**clean Gauss:** Denoised and detrended (DCT 128 sec) with 13 nuisance regressors in GLM + SPM Gauss filter SPM Gauss filter = Gaussian filter set at 2.48 seconds (performed with spm_filter.m)

**SG Pipelines**

**detrend SG (69/6):** Denoised and detrended SG (69/6) with 13 nuisance regressors in GLM

**detrend SG (311/40):** Denoised and detrended SG (311/40) with 13 nuisance regressors in GLM

**clean (15/8):** Denoised and detrended SG (69/6) with 13 nuisance regressors in GLM + SG (15/8) filter

**clean (3/1):** Denoised and detrended SG (311/40) with 13 nuisance regressors in GLM + SG (3/1) filter

In contrast to the optimization experiment, the validation experiment involved assessing the methods by correlating the complete, filtered time courses from both the test and retest experiments. The following metrics were included in the evaluation experiment: Time course reproducibility, Group reproducibility, Connectivity, Connectivity upper bound, Detectable connectivity and Autocorrelation.

### Time course reproducibility

Time course reproducibility was evaluated by correlating filtered, observed time courses from the test and retest runs. We opted to use Pearson correlation over the commonly used Intra Class Correlation (ICC) method for estimating time course reproducibility for the following reasons: I) It is common practice to estimate connectivity with Pearson correlation coefficients and not with ICC. In particular, to remain consistent with current practice, the connectivity upper bound as given in formula (1) should be estimated from test-retest reliability estimates that are based on Pearson correlations rather than ICC. II) fMRI signals are not quantitative in the sense that their height is arbitrary in nature making them less suitable for correlation measures such as ICC that may respond to the absolute height of measurement [22].

We estimated the grand mean and the standard deviations from the 34 time courses*67 subjects = 2278 Pearson correlations. As additional measure for evaluating the reproducibility on a subject level, we computed the percentage of time courses that reached the critical reliability threshold per subject, and averaged these values over the whole sample, which we refer to as the "Mean percentage of time courses within-subjects".

### Group reproducibility

In order to evaluate conventional group reliability, we conducted a conventional ICC (2,1) analysis collapsed over the 67 observations from the initial test run and the 67 observations from the retest run for each individual path [3]. This later process resulted in a total of 561 reliability estimates that were subsequently averaged.

### Connectivity

The connectivity for test and retest runs was calculated for every single individual and subsequently averaged per subject resulting in 561 mean connectivities per subject. We estimated the grand mean and standard deviation of the connectivity from those 561 mean connectivity estimates *67 individuals and report the results in Table 2. Next, we visualized the resulting connectomes with connectome viewer software package (http://nica.net.cn/achievements/t20120530_1132.htm). We used the 3D coordinates of a meta-analysis for this purpose [18]. However, it is explicitly mentioned that even though the connectome appears to be in normal 3D space in reality its alignment took place in 2D spherical aligned space.

## Connectivity upper bound

Note that the connectivity is intrinsically tied to the connectivity upper bound, which is defined as the square root of the product of the test-retest reliability of the time course for node A and node B, represented as ($\rho_{nodeA}$) and ($\rho_{nodeB}$) respectively:

Connectivity upper bound

$$r_{AB} = \sqrt{\rho_{nodeA * \rho_{nodeB}}} \tag{1}$$

We estimated the connectivity upper bound per path per individual.

## Detectable connectivity

The detectable connectivity was calculated as follows:

Observed connectivity was obtained by averaging the connectivity correlations of the test and retest runs per individual.

Observed connectivity was set at zero when one or two nodes exhibited negative or zero reliability and referred to as corrupt connectivity.

If the observed connectivity was positive, the observed connectivity and the connectivity upper bound were compared, and the smaller value was taken.

If the observed connectivity was negative, the absolute value of the observed connectivity and theconnectivity upper bound were compared, and the smaller one was used while making the sign of theresult negative.

Detectable connectivity was averaged for every single path, resulting in the group average connectome consisting of 561 paths. The resulting mean detectable connectivity map was thresholded at r > 0.4 (at least fair) or at r > 0.6 (at least good), as the detectable connectivity map took the underlying test-retest reliability of the path into account. Finally, the grand mean detectable connectivity over paths and participants was estimated and reported in Table 2. The table also shows the percentage of paths that were corrupt or overestimated given the test-retest reliability of the timcourses.

## Autocorrelation

We performed a comparison between the autocorrelation structure of the filtered time courses in the validation experiment and the empirically derived predictor time courses from the optimization experiment. We considered the denoised time course as our ground truth, as the empirical HRF obtained in this way was not subject to any filtering. Utilizing unfiltered data for the predictor time courses is essential when the data is correlated with observed time courses that have been filtered, as it helps avoid problems associated with circularity. To quantify the comparison, we estimated the Root Mean Square Error (RMSE) between the autocorrelations of the predictor time course from the test run and the observed time course from the retest run. We also conducted this comparison vice versa to detect any potential inconsistencies. Results are reported in Table 1.

## Spectral analysis of task and "residual noise time courses"

To ensure that the proposed pipelines could distinguish cognitive signals from noise signals with high accuracy, the power spectra of task and noise time-courses were analyzed using matlab routines. ***Residual noise time-courses*** were obtained by subtracting the SPM or SG cleaned time-courses from the paired denoised and detrended time-courses per subject. Subsequently, every single noise time-course underwent a fast Fourier transform. The resulting power spectra of the Noise SG (3/1), Noise SG (15/8), Noise HRF, and Noise Gauss (2.48 sec) were averaged. The power of the residual noise time-course is expected to be as close to zero as possible in task frequency bands since this indicates that no task signal is removed by the filter.

**Table 1. Comparison of Observed and Predicted Autocorrelations in Test and Retest Run, with RMSE Values.** The grand average autocorrelation is calculated from 34 nodes* 67 individuals. Observed autocorrelations were obtained from various pre-processing methods reported in the left column, predicted autocorrelations were based on denoised data and reported in bottom. was calculated based on the comparison of the observed and predicted autocorrelations from the 4 data points. The pipeline highlighted with a grey background demonstrates an optimal passband characteristic upon validation.

| | observed autocorrelation test run | | | | RMSE from observed versus predicted | observed autocorrelation retest run | | | | RMSE from observed versus predicted |
|---|---|---|---|---|---|---|---|---|---|---|
| | lag 1 | lag 2 | lag 3 | lag 4 | | lag 1 | lag 2 | lag 3 | lag 4 | |
| Raw | 0.66 | 0.48 | 0.36 | 0.24 | 0.17 | 0.66 | 0.48 | 0.37 | 0.24 | 0.18 |
| Denoised (5 wm + 5 ventricle+2 motion components) | 0.55 | 0.34 | 0.21 | 0.09 | 0.14 | 0.58 | 0.37 | 0.24 | 0.12 | 0.15 |
| Denoised Detrend SG (311/40) | 0.51 | 0.25 | 0.09 | -0.05 | 0.20 | 0.51 | 0.24 | 0.08 | -0.06 | 0.19 |
| Denoised Detrend SG (69/6) | 0.51 | 0.25 | 0.10 | -0.04 | 0.19 | 0.52 | 0.25 | 0.09 | -0.05 | 0.19 |
| Denoised Detrend SG (311/40) Filtered SG (3/1) | 0.81 | 0.49 | 0.16 | -0.05 | 0.04 | 0.81 | 0.48 | 0.15 | -0.07 | 0.02 |
| Denoised Detrend SG (69/6) Filtered SG (15/8) | 0.79 | 0.40 | 0.08 | -0.08 | 0.09 | 0.79 | 0.40 | 0.07 | -0.10 | 0.07 |
| Denoised Detrend SPM (128 s.) | 0.54 | 0.31 | 0.17 | 0.05 | 0.15 | 0.56 | 0.33 | 0.20 | 0.07 | 0.16 |
| Denoised Detrend SPM (128 s.) Filter SPM (HRF) | 0.92 | 0.73 | 0.49 | 0.24 | 0.26 | 0.92 | 0.74 | 0.51 | 0.27 | 0.25 |
| Denoised Detrend SPM (128 s.) Filter SPM Gaussian (2.48 s.) | 0.93 | 0.76 | 0.54 | 0.31 | 0.31 | 0.93 | 0.77 | 0.57 | 0.35 | 0.30 |
| | predicted autocorrelation re test run | | | | | predicted autocorrelation test run | | | | |
| | lag 1 | lag 2 | lag 3 | lag 4 | | lag 1 | lag 2 | lag 3 | lag 4 | |
| Denoised | 0.78 | 0.49 | 0.20 | -0.05 | | 0.79 | 0.49 | 0.20 | -0.03 | |

## Monte Carlo simulation

Research has demonstrated that the implementation of filters that remove high frequency noise to clean the data can elevate autocorrelations of time courses, ultimately increasing the deviation of connectivity distributions [12]. To address this concern, we conducted a post hoc Monte Carlo simulation to assess the impact of a specific SG and Gaussian cleaning filter on the standard deviations of the reliability estimates as a function of time course auto correlatedness. The technical description and results are given in S3 Text. Furthermore, we estimated the sample standard deviations of the reliability estimates in real data as they may indicate that some correlations are boosted through the filtering procedure.

## Results

### Results of optimization experiment

The results of the brute force search demonstrated that improper SG filter configuration can severely disrupt the correlation between predicted and observed time courses, while carefully selected SG filter parameters can substantially improve

the correlation. Correlations that were obtained after data were detrended via SG filters were found to range from approximately 0.4 to 0 (see S4-5 Fig). Our data identified the (311/40) SG filter as optimal for detrending, which, however, require a long time series. To determine whether SG detrending could also be effective using shorter time series, we also selected a conservative SG filter (69/6). These filter parameters were chosen since they also produced high correlations between predicted and observed time courses.

Correlations between predictor time courses and time courses where high frequency noise was removed via SG cleaning filters varied dramatically from roughly 0.6 for effective filters down to -0.4 for ineffective ones (S5 Fig). To identify the optimal SG cleaning filter, we minimized the difference between the lag 1–4 autocorrelations of predictor time courses and SG cleaned time courses by calculating the RMSE between the two datasets. The lag 1–4 autocorrelations of the SG cleaned data can be seen in S6 Fig, while the RMSE values resulting from the comparison of the four autocorrelations are displayed in S7 Fig. The filters that met the selection criteria (RMSE < 0.1) belonged to a coherent filter family which SG filter parameters exhibited linear scaling behavior S7 Fig. We selected a cleaning filter of SG (3/1), which equates to a simple moving average, and it demonstrated best performance when combined with an SG (311/40) detrending filter. Additionally, an SG (15/8) cleaning filter showed best performance when paired with a detrending SG (69/6) filter. We decided to compare the moving moving average SG (3/1) filter with a filter of higher polynomial degree as a means to illustrate the drawbacks of moving averages compared to polynomials with higher degrees.

## Validation of autocorrelation structure

In the previous section, we reported on autocorrelations in the context of filter selection. Now, we shift our focus to whether the optimized filters demonstrate suitable autocorrelation behavior when validated using a separate data set. In Table 1, we present the auto-correlations observed in the spatial working memory experiments using various pre-processing methods, along with the auto-correlations of the predicted time-course obtained from the denoised data. The root mean square error (RMSE) between predicted and observed auto-correlations indicates that denoising, SG detrending, and SG cleaning result in the closest match between predicted and observed auto-correlations. This parameter combination outperforms all other pre-processing pipelines (Table 1). Although the use of denoised data as ground truth for the predictor function is reasonable, predicted time-courses based on other pre-processing methods exhibit similar autocorrelation structures, provided that HRF or gaussian filters are not employed (S1 Table). Although the Denoised Detrend SG (311/40) and Filtered SG (3/1) time courses showed slightly better RMSE compared to the Denoised Detrend SG (69/6) and Filtered SG (15/8), we ultimately preferred the latter. This preference was due to its overall autocorrelation structure being lower than the predicted autocorrelation. Which results in a very flat pass-through characteristic of residual noise in cognitively relevant frequency spectra as we will see in the course of the paper.

Table 1 shows that autocorrelation estimates based on HRF and Gaussian filters yield autocorrelations that are too high, while those of raw, cleaned, and SG and SPM detrended time-course sections are too low. It should be noted that denoising and detrending substantially reduces the occurrence of autocorrelations in particular at lag 3 and 4. Additional data cleaning with an SG filter does not further raise the autocorrelations at lag 3 and lag 4.

## Power spectra of signal time courses and time course details

We now examine the impact of different pre-processing procedures on the power spectra and shape of time-courses, see Fig 5.Grand mean power spectra of the working memory time-courses demonstrate that all pre-processing methods maintain the highest power at the task frequency of 0.04 Hz. However, raw time-courses exhibit substantial low frequency noise which is partially removed through denoising with the time-courses of the white matter, ventricles, and head motion. The SG detrending method eliminates additional frequencies below the task frequency that were not removed trough

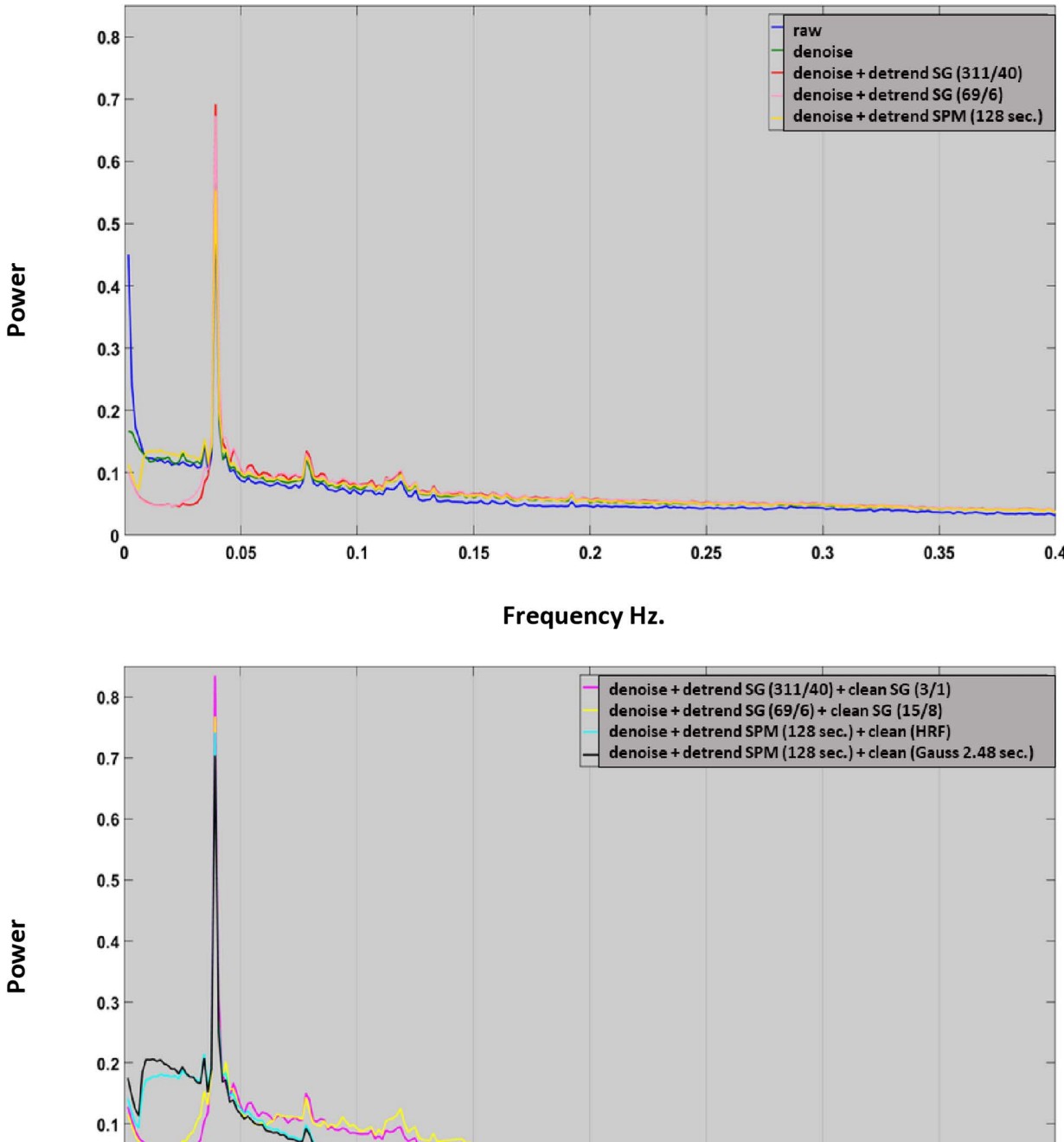

**Fig 5. Effect of pre-processing on the power spectra of working memory related time courses.** The grand mean FFTs were obtained by averaging FFTs over nodes, runs and participants. Pre-processing parameters are given in the legends presented in the right upper corner. Top: the effects of denoising and detrending. Note that SG detrend filters remove frequencies in the 0-0.04 Hz frequency band. Bottom: the effect of the cleaning filters. Note that SPM filters remove more signal in higher frequency bands.

nuisance regression without SG detrending (see Fig 5, top). By contrast, SPM detrending (128 secs. > 0.0078 Hz) shows little effect when combined with denoising methods.

In the following step, we will explore how the effects of various detrending methods, as observed in the FFT analysis, are manifested in the time courses. The top panel of Fig 6 illustrates that the trends identified by SG filters have more intricate shapes at significantly higher frequencies compared to SPM filters. This disparity may explain why SG detrending is more successful in reducing low-frequency noise in comparison to SPM detrending, as noted in the previous FFT analysis (Fig 5 top). The previous FFT analysis suggested that SPM detrending has minimal impact when combined with denoising methods (Fig 5 top), which is supported by the time course data. In the middle panel of Fig 6, we can observe that denoising (green line) eliminates the slow signal trends present in the raw data (blue line), with additional SPM detrending showing little additional effect (red line). Similar observations were found for another exemplary time course (S8 Fig).

A comparison between SPM and SG detrending in the bottom panel of Fig 6 reveals that the time course detrended with an SG filter deviates more from the denoised time course compared to the SPM detrended time courses. Though these differences are subtle, they are sufficiently effective at suppressing low frequency noise not captured by conventional denoising methods.

When examining the cleaning process, Fig 5 illustrates the significant impact of SG and SPM cleaning filters on power spectra. It is evident that the application of SG filters results in a slower decline in power spectra at higher frequencies compared to SPM filters.

The results of this FFT analysis suggest that SPM cleaning may remove task related signal changes in higher frequency bands while this is less the case for SG filters. This can be easily observed when time courses subjected to distinct cleaning methods are compared. Even tough, the SG (3/1) filter is not the optimal cleaning filter as we will discover in the course of this paper it was still able to capture rapid signal changes, which were induced by the complex structure of individual items in the cognitive task (Fig 7 top). By contrast both SPM filters abolished many of the rapid shifts that were induced by the cognitive task (Fig 7 middle). The removal of these rapid cognitive events corresponds with the decrease in power in the 0.04–0.25 Hz frequency band observed in Fig 5 bottom. The destructive effects of the SPM cleaning filters are obvious when they are directly compared to their SG counterparts (Fig 7 bottom). SPM cleaning also left its destructive marks in the empirical HRF. S1 Fig indicated that different preprocessing techniques yielded similar empirical HRF functions as long as cleaning filters were used conservatively. Aggressive cleaning filters disrupted the true shape of the empirical HRF, eliminating relevant information such as signal undershoots etc.

It is concluded that SG filters did not remove rapid signal changes, which were induced by the complex structure of individual items in the cognitive task. In contrast, HRF and Gaussian filters removed these rapid cognitive events, decreasing power in the 0.04–0.25 Hz frequency band.

## Power spectra of residual noise time courses and their relation with the empirical HRF

In the previous section we have observed that in particular the SPM gaussian and SPM HRF filters may remove task related signals from the time course while the latter is less the case for SG filters. However, the prove that the latter is the case is given when the power spectra of residual noise time courses show overlap with true cognitive signal fluctuations. We conducted this analysis by depicting the power spectra of the empirical HRF alongside the power spectra of the residual noise time courses. In Fig 8 A we depict the empirically derived HRF (event-related signal average), which shows minimal signs of noise. Next, we obtain the power spectrum of the empirical HRF, reflecting the true power spectrum of cognitively induced brain responses and depict it in 8 B. We then analyze the power spectra of the residual noise that remains after subtracting the SG or SPM-cleaned time courses from the denoised and detrended time courses and depict the results in Fig 8 C. In a next step we assembled Fig 8 B and 8 C into Fig 8 D. A comparison between the green line (representing the task power spectrum) and the noise spectra reveals substantial differences in the extent to which

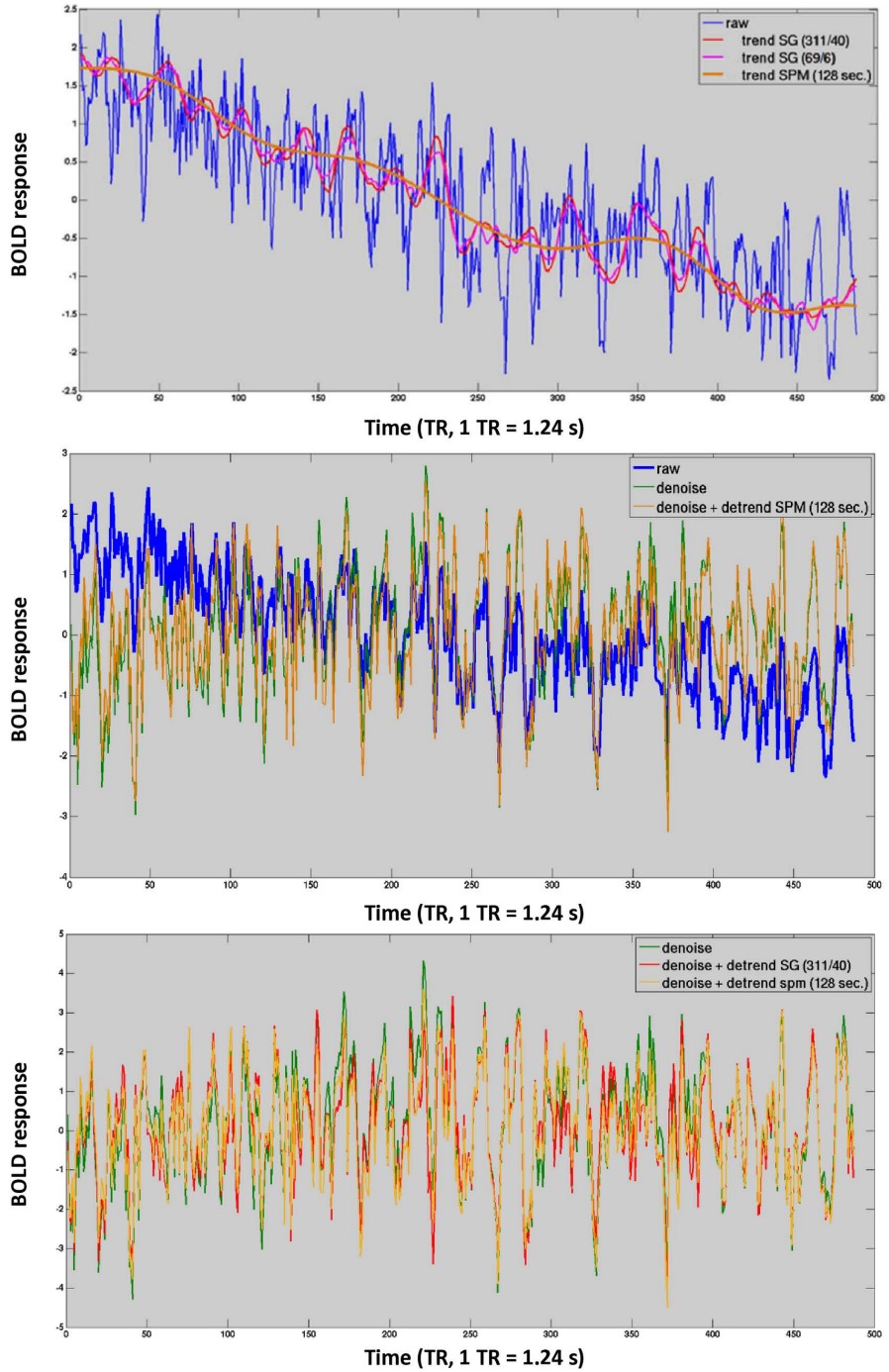

**Fig 6. Effect of detrending on a z-transformed working memory-related time course.** Top: The blue line represents the raw fMRI time course, which exhibits a significant signal trend that requires correction. The conventional SPM filter (brown) captures very slow fluctuations, while the Savitzky-Golay (SG) filters (red and pink) also track micro-trends in frequencies ranging from 0 Hz to 0.04 Hz. Middle: The impact of denoising and simultaneous detrending using the SPM filter (DCT = 128 seconds) is illustrated. The blue line shows the raw time course with the trend, the green line demonstrates how denoising effectively removes much of the slow signal trend, and the red line indicates that the additional benefits of the SPM filter (DCT = 128 seconds) are minimal. Bottom: The effects of denoising and detrending on the time courses of interest for the respective filters, executed within a GLM framework. Note that the SG filter (depicted in red) 'straightens' the signal to a somewhat higher degree than the SPM filter (DCT = 128 sec) depicted in brown.

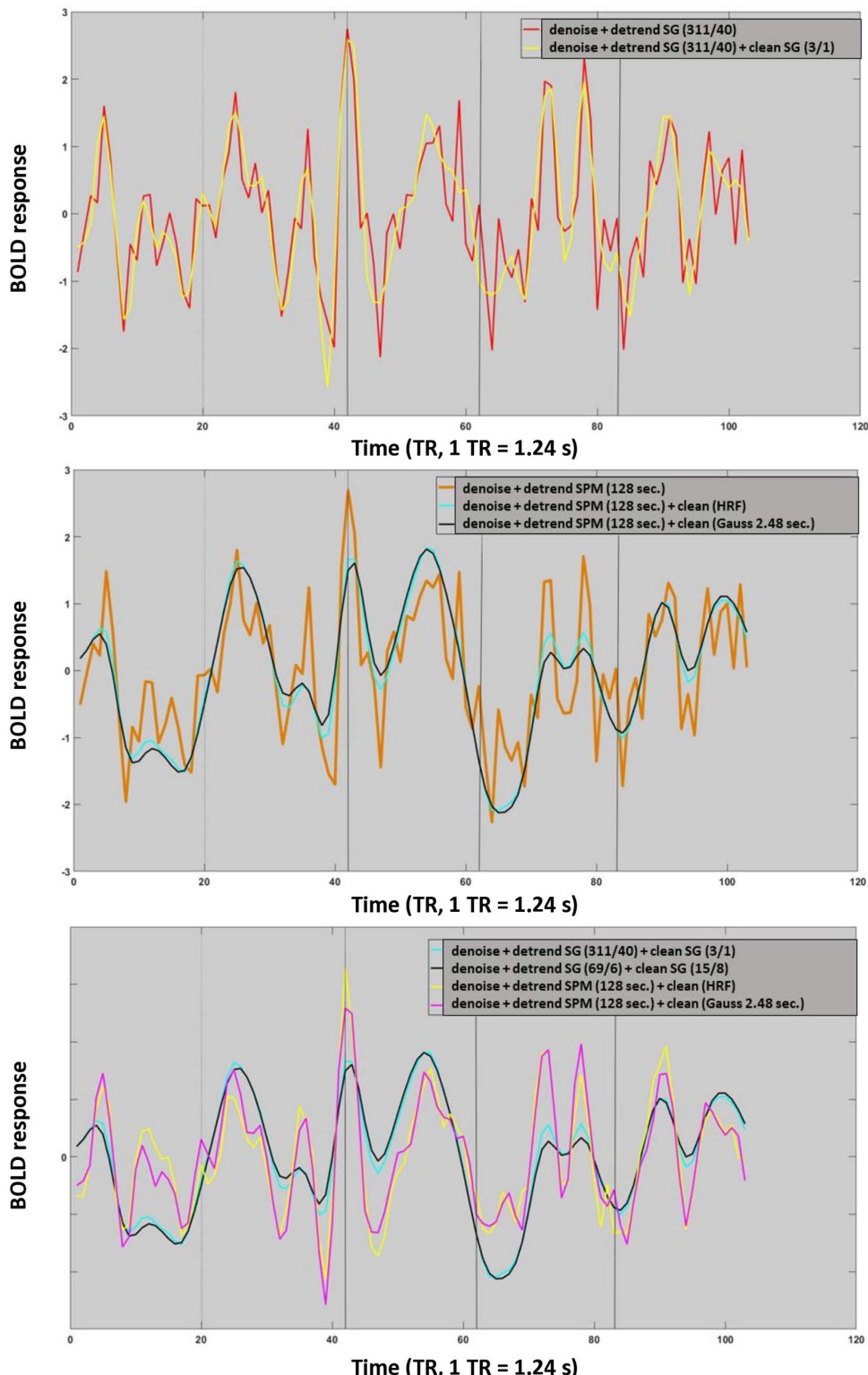

**Fig 7. Effect of cleaning filters on a small fraction of the "working memory task" time-course.** All time-courses were z transformed cleaned and detrended within GLM prior to cleaning. The legend depicted in the right upper corner refer to the preprocessing pipeline in use.

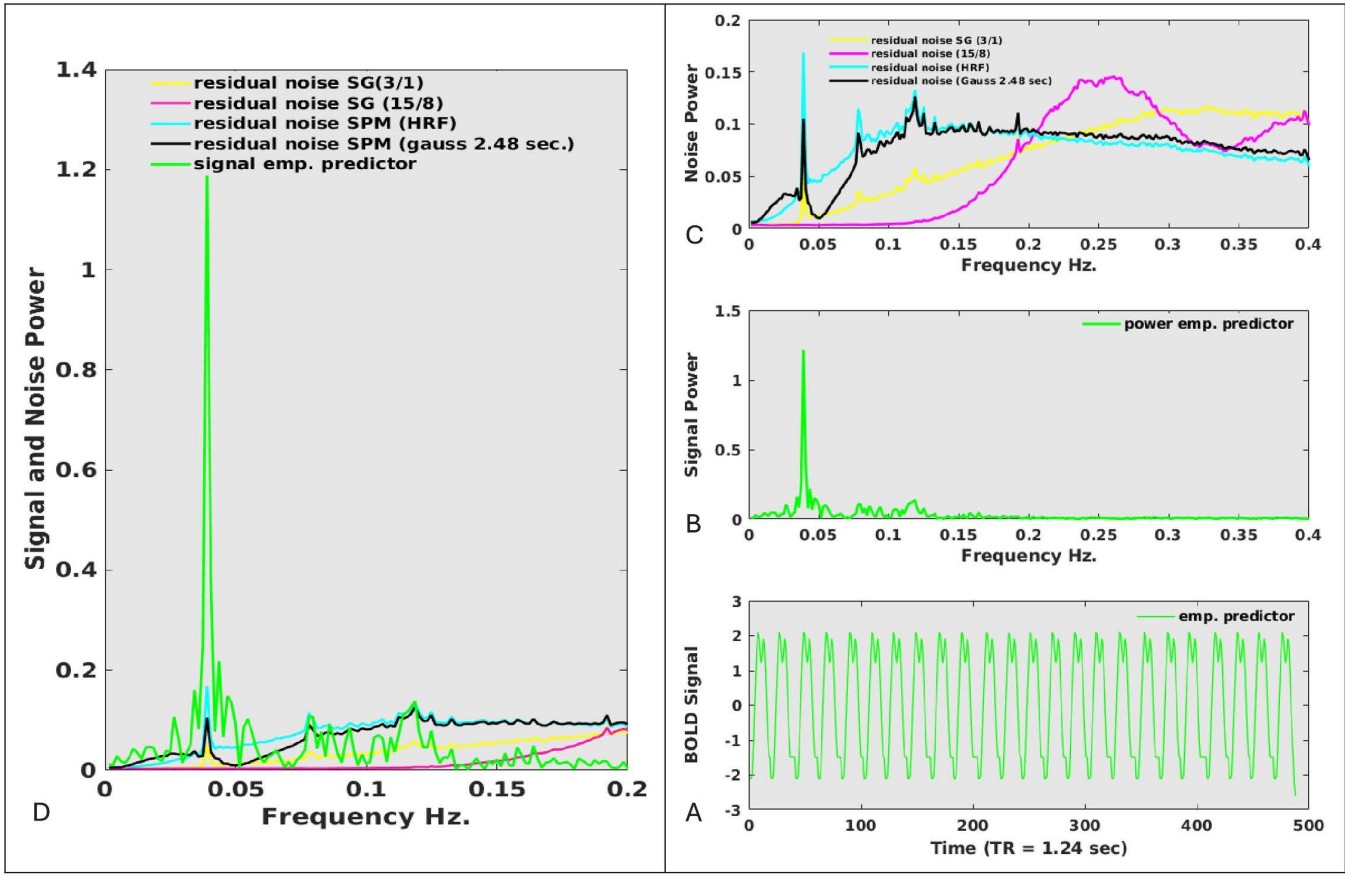

**Fig 8. 8A:** The empirical hemodynamic response function (HRF) was constructed by averaging the individual brain responses across all regions. **8B:** The power spectrum of the empirical HRF presented in Fig 8A. **8C:** Power spectra of the "residual noise time courses." Residual noise was obtained by subtracting the denoised, detrended, and cleaned data from the denoised and detrended data. The legend provides details about the specific pipeline employed. **8D:** Figs 8B and 8C are combined into a single composite figure. It is important to note that the residual noise from traditional SPM filters (Gaussian and HRF) contains significant portions of task signal, whereas this is not observed with the SG(15/8) filter.

the various filters remove the true task signal from the time courses. We observe task-relevant peaks in the residual time courses obtained from SG 3/1 cleaning (low-pass filter) as well as both SPM cleaning filters around 0.04 Hz, which corresponds to the frequency at which the trials were presented. Additionally, we observe task-related brain fluctuations in the ~0.6 Hz to ~0.14 Hz frequency band, which are associated with cognitive shifts occurring within the trial. Importantly, the SG 15/8 cleaning filter does not eliminate signals in task related frequency bands and is therefore in our opinion the filter of choice.

## Reliability. Connectivity, Detectable connectivity

Table 2 shows that denoising has a profound effect on the conventional group reproducibility of paths. While untreated (raw) time courses only exhibited a fair test-retest reliability of ICC = 0.42 denoised time courses exhibited a good test-retest reliability of ICC = 0.70. Adding detrending and cleaning filters did not affect group reproducibility strongly. This suggests that denoising is the most important step of a group analysis while other steps can be omitted. However, the highly optimistic group reproducibility estimates of ICC ~ 0.7 were in sharp contrast with time course reproducibility estimates which

**Table 2. Test-retest reliability and connectivity statistics of working state data on the within-subject level as a function of pre-processing method.** Data were estimated from a connectome that consisted of 34 nodes obtained from 67 participants at two distinct measurement occasions. The pipeline highlighted with a grey background demonstrates an optimal passband characteristic upon validation.

**Summary statistics**

| Within-subject within path reliability | Raw (slice time + Motion correction) | Denoised (5 wm 5 ventricle 2 motion) | Denoised Detrended SG (311/40) | Denoised Detrended SG (69/6) | Denoised Detrended SG (311/40) Cleaned SG (3/1) | Denoised Detrended SG (69/6) Cleaned SG (15/8) | Denoised Detrended SPM (128s.) | Denoised Detrended SPM (128s.) Cleaned SPM (HRF) Filter | Denoised Detrended SPM (128s.) Cleaned SPM(2.48s.) Gauss filter |
|---|---|---|---|---|---|---|---|---|---|
| Group reproducibility ICC(2,1) | 0.42 | 0.70 | 0.73 | 0.73 | 0.73 | 0.73 | 0.73 | 0.70 | 0.69 |
| Grand mean connectivity estimate | 0.44 | 0.43 | 0.47 | 0.46 | 0.59 | 0.54 | 0.44 | 0.58 | 0.57 |
| STD of connectivity | 0.25 | 0.22 | 0.24 | 0.24 | 0.30 | 0.28 | 0.22 | 0.30 | 0.29 |
| Grand mean Time course reliability estimate | 0.28 | 0.25 | 0.35 | 0.32 | 0.48 | 0.41 | 0.26 | 0.41 | 0.38 |
| STD of reliability | 0.23 | 0.16 | 0.20 | 0.19 | 0.25 | 0.22 | 0.16 | 0.25 | 0.23 |
| Grand mean connectivity upper bound | 0.25 | 0.22 | 0.32 | 0.30 | 0.45 | 0.38 | 0.24 | 0.38 | 0.35 |
| Grand mean detectable connectivity | 0.22 | 0.22 | 0.31 | 0.29 | 0.43 | 0.37 | 0.23 | 0.37 | 0.34 |
| % of corrupt paths | 15 | 8 | 3 | 4 | 4 | 4 | 5 | 5 | 6 |
| mean % of overestimated paths | 85 | 93 | 89 | 90 | 85 | 87 | 93 | 89 | 91 |
| mean % of nodes that can be detected r>0.4 | 24 | 15 | 36 | 31 | 65 | 52 | 17 | 50 | 44 |
| mean % of nodes that can be detected r>0.6 | 4 | 1 | 6 | 5 | 26 | 14 | 1 | 16 | 12 |
| mean % of nodes that can be detected r>0.75 | 0 | 0 | 0 | 0 | 3 | 1 | 0 | 2 | 1 |

ranged on average from 0.25 to 0.48 depending on the pipeline in use. Thus, while group reproducibility is not strongly affected by the pipeline of choice - as long as denoising is performed - this is not at the all the case for time course reliability. The main objective of this paper is to assess whether SG filters enhance the detectable connectivity on a within-subject within path level. Specifically, we aim to determine to what extent time-course connectivity is overestimated, considering the underlying test-retest reliability of the time-courses. Results presented in Table 2 demonstrate that the choice of pre-processing strategy significantly influences the height of the detectable connectivity. The data in Table 2 highlight that the reliability and detectable connectivity estimates were roughly identical for untreated (raw) and denoised time-courses, suggesting that denoising does not improve the reliability and connectivity of fMRI time courses. We do not discuss time courses that were processed with SPM or SG (3/1) low-pass filters, as these methods tend to eliminate cognitively relevant aspects of the signal, as demonstrated in Figs 7-8. Additionally, in some cases, they artificially boost the autocorrelation structure

beyond what can be expected based on the empirical predictor. Connectivity, time course reliability and detectable connectivity were substantially improved when timcourses were treated with an SG (15/8) cleaning filter in concert with the full denoising approach. Time course reproducibility was 0.25 for denoised time courses but 0.41 for time courses that were treated with SG filters. This improved the detectable connectivity from 0.22 to 0.37. Therefore, advanced pre-processing techniques have the potential to increase the reliability estimate and the related detectable connectivity very substantially. The histograms in S9 Fig clearly demonstrate the impact of different pre-processing techniques on connectivity and reliability, ultimately influencing detectable connectivity. We observed substantial differences in reliability across various brain regions (see Table S2). The left triangular aspect of the inferior frontal gyrus exhibited the lowest reliability, with a correlation coefficient of $r=0.11$ (raw data), while the middle aspects of the right medial frontal gyrus demonstrated the highest reliability, with $r=0.42$ (raw data). When data were denoised and cleaned with an SG (15/8) filter the reliability of these regions improved to $r=0.21$ for the left inferior frontal gyrus and $r=0.57$ for the right medial frontal gyrus. We also assessed the variability in reliability across individuals for every brain region. Standard deviations ranging from approximately $r=0.10$ to $r=0.25$ suggest that the variability in reliability among individuals is substantial, regardless of the pipeline used (S3 Table).

A formal statistical test revealed that denoising and detrending with the SPM filter does not improve the reliability of the time courses while detrending with SG filters improves reliability in mainly right hemispheric regions (S10 Fig). Applying SG and SPM cleaning filters to the denoised and detrended time courses significantly enhances the reliability of the time courses across all brain regions with t-values up to 30 (S10 Fig).

The increase in detectable connectivity with the number of preprocessing steps was also evident in the connectome images of the SG and SPM pipelines (S11 Fig). The average detectable connectivity should ideally meet the fair reliability criterion. Additionally, considering the high autocorrelations introduced by the SPM pipeline, we limit the discussion of connectomes to pre-processing methods other than SPM. The "raw" image revealed only one fronto-parietal path. The "denoised" image exhibited the right parietal cortex and two frontal nodes. The "detrended" image displayed fronto-parietal connections, as well as an additional network of fronto-parietal connections that emerged primarily in the right parietal cortex. Moreover, frontal connections between the hemispheres were identified. Applying conservative/liberal SG cleaning led to a significant increase in overall connectivity, including connections within the left parietal and frontopolar systems.

### Simultaneous low pass filter and nuisance regression

Previous research on resting-state data suggested that noise is reintroduced into the time-courses of interest when cleaning filters are applied after denoising [21]. It has therefore been advised to develop cleaning filters within a GLM frame work. While the latter may be true for conventional fMRI cleaning filters, we do not advise to develop SG cleaning filters in concert with nuisance regressors since it raises the auto correlations of time courses very substantially and therefore abolishes cognitively relevant aspects of the signal. More details are given in the section S2 Text and S12 Fig and S4-5 Table.

### Does time course cleaning induce correlation?

The Monte Carlo simulation results provided in S3 Text and S13 Fig strongly indicate that the SPM cleaning filter may introduce spurious correlations, whereas this is less the case for SG (15/8) cleaning filters. However, the potentially damaging impact of the cleaning filters on reliability distributions is mitigated when denoising is applied before data cleaning. Comparing the SG/SPM cleaned time courses with the raw time courses reveals that the potentially detrimental effects of the cleaning filters are minimal and not above the original raw data in case of the SG (15/8) filter. This can be interfered from the sample standard deviations of the time course reliability estimates given in Table 2.

### Clinical relevance

We do not discuss time courses that were treated with SPM or SG (3/1) cleaning filter because they remove cognitively relevant aspects of the signal. Instead, we limit ourselves to data that were denoised and detrended with the SPM

cleaning filter, since this reflects the standard case, and data that were detrended SG (69/6) and cleaned SG (15/8) since this reflects the optimal case. The choice of a statistical threshold is always a more or less subjective one and therefore a difficult undertaking. Research has suggested that neural processes may underlie the pareto principle [23]. We applied the pareto principle to the connectome under study meaning that at least 20% of the nodes or paths of the connectome of a single subject should survive a chosen reliability or detectable connectivity threshold. Applying this principle resulted in thresholds of 7 nodes (time courses) or 112 paths respectively. For the cleaned SG (15/8) pipeline in total 94% of the population could reach the threshold of 7 nodes when the threshold was set at test-retest reliability >0.4 while this number was only 37% when time courses were denoised and detrended with a conventional SPM filter. Furthermore, we observed that 87% of the population exhibited at least 112 paths with a detectable connectivity of >0.4 when data were denoised and cleaned SG (15/8) while this number was only 13% for the denoised and SPM detrended pipeline. This suggests that single subject images of scientific quality are feasible in the larger majority of the population when SG filters are used. These figures were substantially lower when the threshold of test-retest reliability or detectable connectivity were set at >0.6. In this case 31% of the population survived the threshold of 7 nodes with r > 0.6 when data were subjected to the SG pipeline while the standard denoised and SPM detrend pipeline resulted in 0 individuals. For detectable connectivity this figure was 10% and 0% for the SG and SPM pipeline respectively. This suggest that at best only one third of the population may profit from fMRI as an additional diagnostic instrument. We give additional information about the effects of preprocessing in Fig 9.

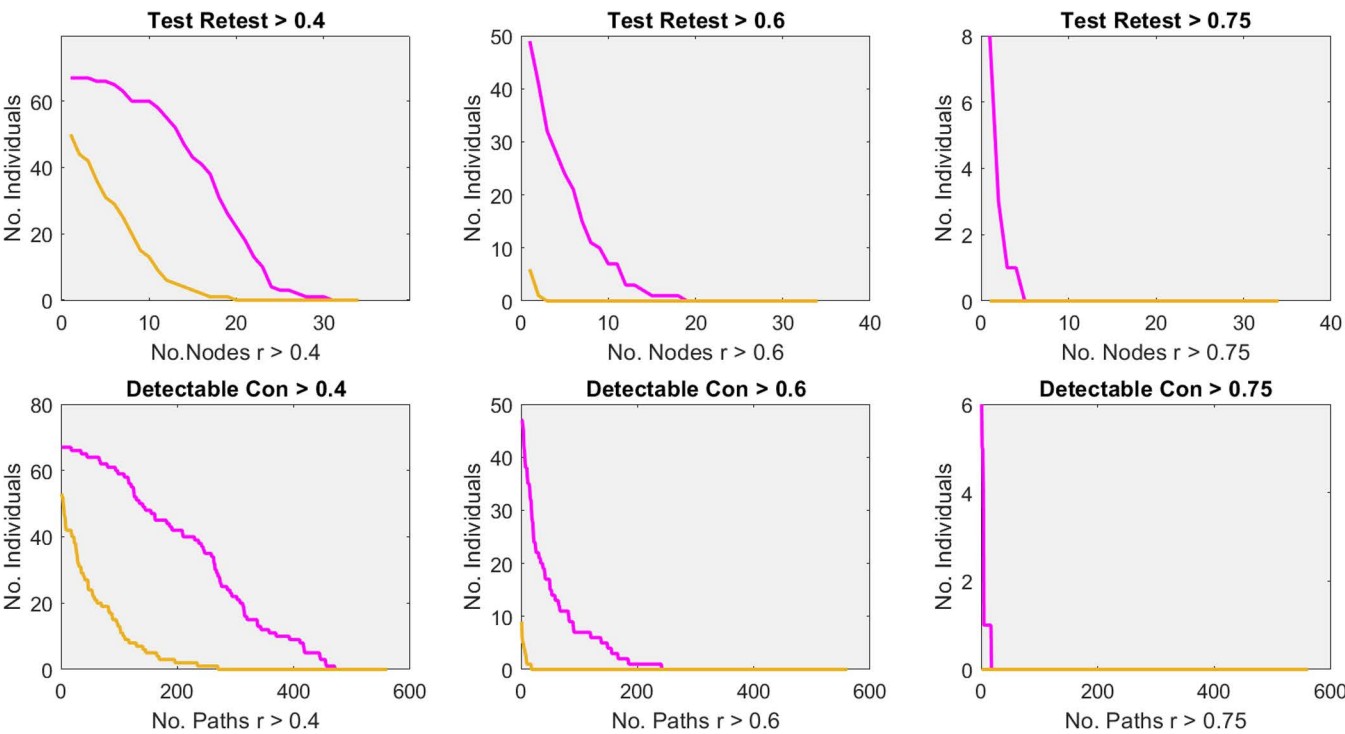

**Fig 9. Reports the effect of the pre-processing pipeline on the number of nodes and paths that could be found for a particular percentage of the sample at a particular correlation threshold.** Pink refers to the pipeline for which data were denoised, detrended SG (69/6) and cleaned SG (15/8) while yellow refers to pipeline for which data were denoised and detrended with the standard SPM filter (128 seconds). Top, reports the percentage of the sample (vertical axis) and the number of nodes (time courses) that reached the critical reliability threshold (horizontal axis). Bottom, reports the percentage of the sample (vertical axis) and the number of paths that reached the critical detectable connectivity threshold (horizontal axis). Note that detectable connectivity implies that test-retest reliability of the underlying sufficiently high.

## Discussion

In this analysis, we demonstrate that the most crucial preprocessing step to enhance conventional group reproducibility in fMRI data is denoising. However, we also find that achieving good group reproducibility does not guarantee that the fMRI images are interpretable for clinical applications, as the within-subject time course reproducibility remains poor at best when modern processing pipelines are employed. Attempts to improve time course reliability using Gaussian and HRF filters, as once employed in SPM, removed rapid cognitive shifts from the time course, as depicted in Fig 8. This removal of cognitive information resulted in highly smooth time course progressions as shown in Fig 7. While these filters improved time course reliability, it is important to note that they also altered the autocorrelation structures within the time courses. Our computer simulation, shown in S13 Fig, indeed demonstrated that the improvements in time course reliability are not genuine, as they were caused by increases in autocorrelation. A promising approach to improve the clinical applicability of fMRI images is to implement refined filtering techniques that conserve the cognitive aspects of the signals. By focusing on preserving the meaningful components of the signal while reducing noise, these advanced filters can enhance the interpretability of the data for clinical purposes. SG filter-based pipelines accurately extract cognitive signals from noisy backgrounds which remains challenging for other cleaning pipelines [24]. It should be noticed that while SG filters do not remove true cognitive signals, they may not always remove true noise. This raises a dilemma: should one prioritize minimizing the risk of rejecting true noise, or should one focus on minimizing the risk of rejecting true cognitive signals? We have chosen the latter option, as we believe it represents the most conservative approach. However, in ultimo ratio the balance between "Type I error" and "Type II error" is based on the research assumptions of a scientist.

### Oscillations in frequencies between 0 Hz and 0.04 Hz

SG detrending methods detect non-task-related fluctuations in the BOLD response occurring in the resting state frequency band, which are substantially faster than fluctuations removed through conventional detrend filters. We have no definitive explanation for the origin of this activity. One explanation is that SG detrend filters isolate physiological noise, which is not isolated by conventional denoising pipelines. Since 12 noise regressors were used to remove physiological noise, other explanations are possible as well. One might speculate that non-task-related (resting state) brain activity leaks into the actual signal of cognitive interest. As for now, oscillations in frequencies between 0 Hz and 0.04 Hz seem to occur in the underexplored realm between noise and cognition. We would like to call these oscillations zombie artifacts. Zombie artifacts, along with other MRI phenomena such as Ghosting artifacts [25], Magic angles [26], Voodoo-correlations [27], Herringbone artifacts, Zebra artifacts, and Popcorn artifacts [28], should be subject to appropriate control measures as they may corrupt reproducibility.

### Detectable brain connectivity

A previous study reported that test-retest reliability of time-courses that were pre-processed with conventional methods may not exceed $r = 0.25$ [4]. This suggests that detectable connectivity among region A and B cannot exceed $r = 0.25$. We observed detectable connectivity values around $r = 0.23$ for time courses that were denoised and detrended with a standard SPM filter which are very close to the previously reported estimates. This is somewhat puzzling since connectivity in this network was estimated at 0.44 in the present study, meaning that roughly half of the observed connectivity is false [6]. Ideally, functional connectivity pipelines should maximize detectable connectivity while minimizing spurious autocorrelations. Our analysis shows that SG filters are a step in the right direction in particular when they are compared with denoise filters that were once standard in the SPM package. In defense of the SPM package, we would like to point out that cleaning filters have been removed from the package since SPM 5. We do not exude the possibility that gaussian or HRF filters may enhance reliability when parameters are optimized for current purposes. SG filters that are used to remove high frequency noise maximize detectable connectivity of time-courses while maintaining realistic autocorrelation levels. Furthermore, spectral analysis of "residual noise time courses" revealed that SG (15/8) filters did not remove task related signals while this was not the case for other filter types. Grand mean detectable connectivity of a working memory

related connectome was r = 0.37 for the optimal SG pipeline as opposed to a detectable connectivity of around r = 0.23 for time course that were denoised and detrended with a standard SPM filter. One might argue that brain activity is constantly fluctuating and changing, even in the presence of repeating, external stimuli. Therefore, there are natural limits to the amount of reliability that can be detected. Recently we showed that within-subject test-retest reliability of response behavior is correlated with within-subject reliability of brain responses [5]. This suggests that individuals who display unreliable behavioral responses may also exhibit inconsistent brain responses. This observation holds clinical significance, as some individuals may be inherently unsuitable for the clinical applications of fMRI, as they cannot control within-subject signal fluctuations. Therefore, it is crucial to assess the test-retest reliability of patients undergoing pre surgical mapping.

### Cleaning filters within GLM framework

Previous resting state studies claim that overestimation of connectivity is reduced when filters and nuisance factors are integrated within a GLM framework [21]. Our analyses revealed that detectable connectivity is boosted when SG low-pass filters and other nuisance factors are regressed out simultaneously. However, the price of this operation is high, for it leads to a substantial increase in temporal autocorrelations indicating that task-related signals were removed.

### Autocorrelation behavior

In our analysis, we found that denoising and detrending techniques effectively reduce the presence of serial autocorrelations, thereby significantly mitigating the risk of biased connectivity and reliability estimates. Moreover, the use of SG cleaning filters does not exacerbate autocorrelations at lag 3 and 4, indicating that the overall SG filtering procedure can potentially improve temporal resolution while reducing noise levels.

### Clinical feasibility of fc-MRI

Our analysis has revealed that the high conventional group reproducibility (r ~ 0.7) of single paths can be misleading in the context of clinical fMRI, which inherently focuses on the time course reproducibility of a single subject. Unfortunately, mainstream pipelines have shown disappointing time course reproducibility, resulting in poor detectable connectivity. Therefore, we restrict our discussion to data that were denoised and cleaned with a SG (15/8). Currently single subject connectomes of scientific quality (r > 0.4) can be obtained in 94% of the sample when SG pipelines are used while this number is 37% for main stream pipelines. But our findings also showed that approximately only one third of the sample demonstrated good reproducibility at the level of single nodes (time course reliability >0.6), while this figure was only 10 percent for connectivity measures (detectable connectivity>0.6). It is important to understand that there is a natural hierarchy when it comes to the reproducibility of different fMRI methods. Specifically, in terms of time course reproducibility, the level of reproducibility is influenced by the noise levels present in the two-time courses that are being examined. On the other hand, in the context of connectivity, the reproducibility is constrained by the test-retest reliability of the two-time courses being investigated. Therefore, the small fraction of the sample with sufficient detectable connectivity (r > 0.6) is to be expected given this hierarchy. It should be noticed that even though a test-retest reliability of r > 0.6 seems to be high this may only be sufficient for fMRI as an additional diagnostic tool in the context of well-established gold standard practices. Recent research around the applicability of fMRI in the context pre surgical mapping reported sensitivity ranging from 59% to 100% and specificity from 0% to 97% [8,9]. It is very well possible that the thresholds we have chosen in this study are overly conservative and may need adjustment when the proposed pipeline is used in pre-surgical mapping.

### Standardization of software packages

Critics of our approach may argue that yet another pipeline is added to the very large family of pipelines available, and that this wide variety of pipelines hinders the comparative interpretation of neuro imaging results that have been obtained

with distinct alignment methods, smoothing kernels, temporal filters, interpolation methods, machine precision, and randomization methods [21,29,30]. The lack of standardization may indicate that the field of bold imaging has not yet fully matured, as there is no generally accepted standard pipeline A fate that is shared with other developing fields such as genome wide association [31–35]. However, with the emergence of systems like fMRIprep, it is now feasible to combine the best of all worlds. In this context, advanced band-pass filters can be integrated with the various nuisance regressors that this system provides.

### Limitations of this study

It is possible that classic Gaussian low pass filters and ICA driven cleaning may outperform SG filters when optimized for specific experimental settings. Our study's reproducibility estimates are somewhat higher than those reported by Gorgolewski [4], but the reliability of our pipeline may decrease when measurement occasions are further apart [36]. Another limitation is that we did not remove residual autocorrelations from the time courses before estimating time course reliability, potentially compromising reliability estimates. SG based pipelines still exhibit high noise levels between task peaks, possibly due to inherent individual differences in task behavior and BOLD expression [37]. It is challenging to differentiate neural or physiological noise from neuro psychological response variability within subjects. We suspect that physiological or other sources of unwanted noise remain in the 0.05–0.2 Hz frequency band after SG filtering, as the power spectrum analysis did not approach zero between task-related frequency peaks. This remaining noise is a result of the conservative filter policy aimed at minimizing the risk of removing true signals. We did not obtain additional sources of physiological/psychological noise, such as heart rate or respiration rate, which could have aided in interpreting the spectral analysis. Recent research suggests that noise measures from physiological sensors differ from those obtained from the brain itself [38], highlighting the importance of both forms of cleaning. It was beyond the reach of this study to correct for serial correlations and we speculate that the frequently used AR(1) model may not be suitable when time courses have been treated with SG filters as the low auto correlations found at lag 3 and 4 after filtering suggest that correlations removal should be limited to lag 1 and 2 [39]. However,it was beyond the reach of this study to build a complex ARIMA model that does justice to this specific auto correlation structure. Although a lot of the autocorrelations have been removed by the SG pipeline it may still induce spurious correlations albeit to a lower degree when compared to conventional pipelines. Alternatively, one might argue that such corrections are not strictly needed as higher auto correlations at lag 1 and 2 possible reflect a genuine cognitive process. Our sample includes a broad range of individuals including older people. However, we do not know how well our pipeline performs in actual patients that may suffer from traumata that may further corrupt the reproducibly of fMRI. Finally, our pipelines were developed for a specific cognitive task that was sampled with a specific repetition time and voxel size. It is very likely that our pipeline does not generalize to other setting. But current research suggests that the time in which one pipeline fits all purposes is coming to an end [40].

### Conclusions

Improving the test retest of single subject images is of immediate relevance for clinical applications of fMRI. It may be beneficial to design optimal pipelines for specific machines and clinical purposes that mainly aim at the detection of Broca's area Wernicke's area and the hand area. The road to fMRI as a diagnostic standalone tool may be covered with stones although we speculate that future system might distil cognitive information from highly noisy time courses by means of artificial intelligence [41]. Using very stringent criteria we observed that for roughly 10–30% of the population fMRI may be a meaningful diagnostic tool (time course reproducibility > 0.6) when advanced filters were used while for conventional approaches this figure was nihil.

## Supporting Information

**S1 Code.**
(ZIP)

**S2 File.**
(PDF)

## Author contributions

**Conceptualization:** Jan Willem Koten, André Schüppen.

**Data curation:** Jan Willem Koten, Guilherme Wood.

**Formal analysis:** Jan Willem Koten.

**Funding acquisition:** Guilherme Wood.

**Investigation:** Jan Willem Koten.

**Methodology:** Jan Willem Koten, André Schüppen, Martin Holler.

**Project administration:** Guilherme Wood.

**Software:** Jan Willem Koten, André Schüppen.

**Validation:** Jan Willem Koten.

**Visualization:** Jan Willem Koten.

**Writing – original draft:** Jan Willem Koten, André Schüppen, Martin Holler.

**Writing – review & editing:** Jan Willem Koten, Martin Holler.

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
