## [Decision Letter · Decision Letter 0]

18 Dec 2024

PONE-D-24-38474Towards clinical applicability of fMRI via systematic filteringPLOS ONE

Dear Dr. Koten,

Thank you for submitting your manuscript to PLOS ONE. After careful consideration, we feel that it has merit but does not fully meet PLOS ONE’s publication criteria as it currently stands. Therefore, we invite you to submit a revised version of the manuscript that addresses the points raised during the review process.

We look forward to receiving your revised manuscript.

Kind regards,

Kendrick Kay

Academic Editor

PLOS ONE

Journal Requirements:

“This study was supported by FWF grant (P 22577-B18).”

Reviewers' comments:

Reviewer's Responses to Questions

**Comments to the Author**

1. Is the manuscript technically sound, and do the data support the conclusions?

Reviewer #1: Partly

Reviewer #2: Yes

2. Has the statistical analysis been performed appropriately and rigorously? 

Reviewer #1: No

Reviewer #2: Yes

3. Have the authors made all data underlying the findings in their manuscript fully available?

Reviewer #1: Yes

Reviewer #2: Yes

4. Is the manuscript presented in an intelligible fashion and written in standard English?

Reviewer #1: Yes

Reviewer #2: Yes

5. Review Comments to the Author

Reviewer #1: Koten and colleagues performed a well-thought-out and clearly explained set of experiments to test whether a new set of detrending and denoising methods could improve the reproducibility of task-based fMRI over conventional detrending and denoising methods. The authors also used a clever validation framework that allowed them to avoid bias in their methodology. However, the results are not presented very clearly, and I believe there are issues with the results and interpretations that need to be addressed before this article is ready for publication, as outlined below.

I am not convinced that the SG filters improve the reliability of fMRI signals or connectivity over the standard SPM methods based on these data. Although it is allued to that SG filters are more reliable, the “SPM denoised, detrended, cleaned, HRF” pipeline seems to lead to just as good reliability as the SG pipelines. For instance, the time course reliability of the SPM pipeline is 0.41, while the time course reliabilities of the best two SG pipelines are 0.48 and 0.41. If the claim is to be made that the SG pipelines have better reliability, there needs to be either a statistical comparison, or a much more substantial difference in reliability. Additionally, why is the one pipeline highlighted red in Tables 1 and 2? This is not obviously the best performing pipeline in each table to me. It should be noted somewhere why this pipeline is highlighted, and if this pipeline is considered “optimal”, the reasoning should be noted somewhere.

Furthermore, the reliability measures of time courses in Table 2 are grand means, how do these reliability measures vary across participants and regions? Could these measures be statistically compared between pipelines if the grand mean is not calculated?

Figure 9 shows that the SG method produces better connectivity reliability than the SPM method. However, based on Table 2, the two SPM methods that were tested, but not presented in Figure 9, look as if they have as high or near to as high connectivity as the SG methods. For a fair comparison of SG to SPM methods, either all methods or the methods with the highest reliabilities should be compared in Figure 9.

In the section on power spectra of signal time courses, why is it assumed that the SG filters retaining high frequency signal, but removing low frequency signal is akin to removing unwanted noise? I would imagine that much of this high frequency signal is also due to unwanted noise and is not solely task-related. The signals should be compared to the predictor time courses using ICCs to determine which detrending/cleaning method is optimal, as is done later in the manuscript.

All figure axes should be labeled with units. It’s hard to evaluate the results in the figures without this. As an example, I cannot tell what the time units are for Figure 6, so I do not know how to convert these to frequency as mentioned in the text.

This study seems to assume that reproducibility of 1 is the gold standard. However, we know that brain activity is constantly fluctuating and changing, even in the presence of repeating, external stimuli. Therefore, some variance between sessions should be expected in the fMRI signal, even in the presence of absolutely no noise.

If the gold standard time course is the predictor time course, shouldn’t this be plotted in Figures 6 and 7 so that we can visually evaluate which pipeline is the closest to what we would expect?

Figure 6: I think the top figure plots the “residual” time courses, correct? If so, this should be mentioned since it is unclear as written.

It seems like the individual HRF and “event-related average” are used somewhat interchangeably. This seems correct to me, but it would be clearer if one term was defined and used.

Reviewer #2: Reproducibility assessment of fMRI is a common practice at group level. Conventional signal post processing of fMRI has shown poor reproducibility on an individual level. The study aims to introduce a data driven signal filtering workflow with Savitzky-Golay filters to fMRI signal for improving single subject reproducibility in comparison SPM’s default signal filtering method. By improving subject level reproducibility, the work aims to improve the potential value of fMRI scans in clinical assessments.

Overall the experiments are documented in great detail and comprehensive. However the current workflow is focused on SPM based workflow and described in some SPM specific terms. To allow the research to have a wider reach in the general fMRI research community, I have some suggestions to allow this work to be adopted by FSL, fMRIPrep, and AFNI users. It was a really good read and I recommend the study to be published in PLOS One after addressing the several points listed below:

SPM filter: please elaborate the nature of this filtering method. SPM uses a 128s cutoff for high pass filter by default, and creates cosine regressors. This is identical to fMRIPrep’s recommendation, and can be implemented in other types of workflows.

GLM frameworks for deriving nuisance regressors: PCA was done on nuisance regressors of each category (white matter, CSF, motion). There’s no description of what the regressors are in detail to derive the principal components. Can you please elaborate:

Moton: are you using just the the 6 rigid-body motion parameters (3 translations and 3 rotation), or included temporal derivatives and quadratic terms (described in Satterthwaite 2013; 6 base motion parameters + 6 temporal derivatives of six motion parameters + 12 quadratic terms of six motion parameters and their six temporal derivatives=24 regressors)?

White matter and CSF signals: same issue applies temporal derivatives and quadratic terms applies (1 base parameters + 1 temporal derivatives of base parameters + 2 quadratic terms of base parameter and their temporal derivative=4 regressors for WM and CSF each). Or is this derived by compcor (Behzadi2007) as the number of regressors of WM and CSF are 5?

I would recommend to consult the fMRIPrep documentation (https://fmriprep.org/en/stable/outputs.html#confound-regressors-description) and BIDS functional derivative (https://bids-specification.readthedocs.io/en/bep012/derivatives/functional-derivatives.html) for detailed descriptions.

Data driven denoising regressors (such as CompCor and ICA-AROMA) have shown inconsistency between software versions, hence hinders reproducibility. The authors used a fixed number of PC regressors to ensure the same loss of temporal degrees of freedom across subjects, but this doesn’t prevent the workflow from introducing noise through float point differences (see Compcor 6 and aroma in Fig 6 vs Fig 11. https://doi.org/10.1371/journal.pcbi.1011942). Have the authors fixed the random seed to ensure the reproducibility of the workflow within the scope of the research project? The reason can beI would like the authors to address this as a limitation of the current workflow.

Following point 3, I am curious to see the impact of SG filter on other workflows established in the literature (see https://doi.org/10.1016/j.neuroimage.2017.03.020 and https://doi.org/10.1371/journal.pcbi.1011942). However this is not crucial to the manuscript, since the comparison listed in the study is sufficient to highlight the effect of SG filter.

6. PLOS authors have the option to publish the peer review history of their article (what does this mean? ). If published, this will include your full peer review and any attached files.

**Do you want your identity to be public for this peer review?** For information about this choice, including consent withdrawal, please see our Privacy Policy .

Reviewer #1: No

Reviewer #2: No

---

## [Author Response · Author response to Decision Letter 0]

17 Feb 2025

5. Review Comments to the Author

Reviewer #1: Koten and colleagues performed a well-thought-out and clearly explained set of experiments to test whether a new set of detrending and denoising methods could improve the reproducibility of task-based fMRI over conventional detrending and denoising methods. The authors also used a clever validation framework that allowed them to avoid bias in their methodology. However, the results are not presented very clearly, and I believe there are issues with the results and interpretations that need to be addressed before this article is ready for publication, as outlined below.

I am not convinced that the SG filters improve the reliability of fMRI signals or connectivity over the standard SPM methods based on these data. Although it is allued to that SG filters are more reliable, the “SPM denoised, detrended, cleaned, HRF” pipeline seems to lead to just as good reliability as the SG pipelines. For instance, the time course reliability of the SPM pipeline is 0.41, while the time course reliabilities of the best two SG pipelines are 0.48 and 0.41.

We appreciate the reviewer’s insightful comments regarding the test-retest reliability of the SPM denoised, detrended, cleaned (HRF filter) in comparison to the proposed SG pipeline. We acknowledge that the SPM pipeline might indeed perform comparably to our SG pipeline when its Gaussian low-pass filter parameters are optimized for contemporary applications, as we have previously noted in the limitations section of our paper. However, it is important to highlight that the current parameter settings of the SPM pipeline exhibit certain shortcomings that cannot be overlooked. Specifically, the SPM low-pass filters significantly increase the temporal autocorrelations compared to the SG low-pass filters. For instance, the temporal autocorrelations of the SPM denoised, detrended, cleaned (HRF filter) signal are 0.92, 0.73, 0.49, and 0.24 for lags 1 to 4 in the test run. In contrast, the temporal autocorrelations for the Denoised; Detrend SG (69/6); Filtered SG (15/8) pipeline are 0.79, 0.40, 0.08, and -0.08. It has been noted that artificially induced autocorrelations may cause spurious correlations [1]. The latter may affect both connectivity and test-retest reliability. Given the high autocorrelations of the signal after SPM low-pass filtering, one may wonder if these improvements in reliability are genuine [1]. Our simulation depicted in Figure S13 indeed shows how particularly SPM filters artificially increase test-retest reliability through higher autocorrelations. We propose that the ‘filtered signal cannot be better than the original signal,’ which we define as the autocorrelational structure of the empirical HRF obtained from a different dataset to avoid circularity. The autocorrelational structure of the empirical HRF is 0.78, 0.49, 0.20, and -0.05. Notably, the autocorrelations of the selected SG filter are very close to, or even below, those of the empirical HRF, whereas the SPM pipeline exhibits autocorrelations that exceed those of the empirical HRF. We believe our approach has merit, as illustrated in Figure 7, which clearly demonstrates that the signals treated with the SG filters closely resemble the signals that were not low-pass filtered. In contrast, the SPM filters tend to alter the signal substantially. This is particularly evident when examining the power spectra of the residuals remaining after filtering, as shown in the novel Figure 8D. While the Denoised; Detrend SG (69/6); Filtered SG (15/8) pipeline does not exhibit peaks in task-relevant frequency bands (0.00001-0.15 Hz), the other filters do remove cognitively relevant aspects of the signal. More specifically, the frequency peak at ~0.04 Hz observed in Figure 5 is most likely connected to the trial-related signal shifts that are induced by the cognitive tasks. The very same peak dominates the residual noise in Figure 8. For this reason, along with others, the use of SPM low-pass filters, as established in SPM99 (https://www.fil.ion.ucl.ac.uk/spm/doc/manual/manual.pdf), has been criticized [2]. In the subsequent SPM5, the low-pass filters were removed in favor of the AR(1) model (https://www.fil.ion.ucl.ac.uk/spm/doc/spm5_manual.pdf). For the very same reason, we do not favor an SG low-pass filter (105/35) that was integrated within the GLM, although it clearly outperforms all other pipelines in terms of test-retest reliability (r=0.53). We finally selected the Denoised; Detrend SG (69/6); Filtered SG (15/8) pipeline because the pass-through characteristic of residual noise depicted in Figure 8D is very flat in cognitively relevant frequency spectra, making it, at least for this experiment, the optimal choice as it improves reliability while at the same time keeping the cognitive aspects of the signal intact.

If the claim is to be made that the SG pipelines have better reliability, there needs to be either a statistical comparison, or a much more substantial difference in reliability. Additionally, why is the one pipeline highlighted red in Tables 1 and 2? This is not obviously the best performing pipeline in each table to me. It should be noted somewhere why this pipeline is highlighted, and if this pipeline is considered “optimal”, the reasoning should be noted somewhere.

We have now added a new Figure S10 in which differences in test-retest reliability between all the pipelines are assessed using a repeated measures t-test. The pipeline highlighted in red in Tables 1 and 2 always corresponds to the SG-based pipeline. We have added an additional comment in the figure caption to clarify this. Regarding the reliability of the SG pipelines, we refer to our response to the previous question: Given the significant shortcomings of the other methods as described above, we believe it is sufficient that the reliabilities of the proposed SG pipeline is comparable to that of the other methods.

Furthermore, the reliability measures of time courses in Table 2 are grand means, how do these reliability measures vary across participants and regions? Could these measures be statistically compared between pipelines if the grand mean is not calculated?

We agree with the reviewer that this is an important point. We have now added an extra supporting material (Figure S10) that reports on the statistical differences between the pipelines by means of a repeated measure t test for which the resulting p values were corrected for multiple comparisons. However, we believe that the most appropriate comparison between our SG pipeline and the SPM pipeline is one in which the SPM pipeline excludes both the HRF filter and the Gaussian filter. This is because the authors of SPM no longer consider these filters to be valid. As mentioned earlier, these filters can introduce high autocorrelations in the time series, potentially leading to spurious correlations and the removal of genuine task signals [1]. In addition, we have added Table S2, which provides the mean values for each brain region across all pipelines, as well as the standard deviation for each region per pipeline. Table S3 details how reliability varies across subjects.

Figure 9 shows that the SG method produces better connectivity reliability than the SPM method. However, based on Table 2, the two SPM methods that were tested, but not presented in Figure 9, look as if they have as high or near to as high connectivity as the SG methods. For a fair comparison of SG to SPM methods, either all methods or the methods with the highest reliabilities should be compared in Figure 9.

We have now added these figures, however they are now referred to as S11 Figure. SPM Connectomes

In the section on power spectra of signal time courses, why is it assumed that the SG filters retaining high frequency signal, but removing low frequency signal is akin to removing unwanted noise? I would imagine that much of this high frequency signal is also due to unwanted noise and is not solely task-related. The signals should be compared to the predictor time courses using ICCs to determine which detrending/cleaning method is optimal, as is done later in the manuscript.

We agree to this comment. In fact, this is how the optimal SG filters were found during the optimization procedure described . That is, we correlated the preprocessed (filtered) time course of a test run with the empirical HRF that was constructed from task related signal averages from a retest run to avoid circularity. The whole procedure has been visualized in Figure S2 and S3. Subsequently only SG filters that showed high correlation with the predictor function and also showed the same auto correlational structure as the predictor function were retained. The SG filters that were potentially suitable are visualized in Figure S5-7.

However, the basic skepticism of the reviewer is nonetheless appropriate as the reviewer wants to see prove that the SG 3/1 low pass filter as well as both SPM filters remove cognitively relevant aspects of the signal during the actual test phase of experiment. As pointed out in the method section ‘time course reliability’ fMRI signals are not quantitative in the sense that their height is arbitrary in nature making them less suitable for correlation measures such as ICC that may respond to the absolute height of measurement.

We have approached the problem from a different angle and hope that the reviewer appreciates our efforts. In a novel Figure 8, we depict the empirically derived HRF (event-related signal average), which shows minimal signs of noise. Next, we obtain the power spectrum of the empirical HRF, reflecting the true power spectrum of cognitively induced brain responses, and present it in Figure 8B. We then analyze the power spectra of the residual noise that remains after subtracting the SG or SPM-cleaned time courses from the denoised and detrended time courses in Figure 8C. If the power spectra of the residual noise exhibit peaks in task-relevant frequency bands, it suggests that relevant signals were removed by the method. This is further evidenced when the power spectra of the residual noise time courses overlap with the power spectra of the empirically derived HRF.In Figure 8D, we indeed observe task-relevant peaks in the residual time courses obtained from SG 3/1 cleaning (low-pass filter) as well as both SPM cleaning filters around 0.04 Hz, which corresponds to the frequency at which the trial was presented. Additionally, we observe task-related brain fluctuations in the ~0.6 Hz to ~0.14 Hz frequency band, which are associated with cognitive shifts occurring within the trial. Importantly, the SG 15/8 cleaning filter does not eliminate any task signal.

Returning to the reviewer’s argument, we acknowledge that much of this high-frequency signal may also stem from unwanted noise and might not solely be task-related. The reviewer is correct in noting that the SG 15/8 filter does not remove all unwanted noise. This raises a dilemma: should one prioritize minimizing the risk of rejecting true noise, or should one focus on minimizing the risk of rejecting true cognitive signals? We have chosen the latter option, as we believe it represents the most conservative approach.

We now also mention this dilemma in the discussion of the paper.

All figure axes should be labeled with units. It’s hard to evaluate the results in the figures without this. As an example, I cannot tell what the time units are for Figure 6, so I do not know how to convert these to frequency as mentioned in the text.

We have now labeled the axis.

This study seems to assume that reproducibility of 1 is the gold standard.

We do not believe that the gold standard is a correlation of 1; rather, we consider the empirically derived hemodynamic response function (HRF) to be the true gold standard. Nonetheless, fMRI should exhibit excellent within subject test-retest reliability for clinical applications. According to Cicchetti, this is the case if the ICC > 0.75 while Li and Koo require ICC > 0.9. Therefore, it is essential to focus on improving the reproducibility of fMRI analyses, as this is crucial to gain the acceptance of the method among clinicians, including neuropsychologists, neuroradiologists, and neurologists etc. While we do not reach these levels of test-retest reliability with our pipeline we can nonetheless improve the test reliability such that at least some individuals can be scanned with r>0.6. A value of at least 0.6 is considered good reproducibility, which may allow fMRI to be used as an “ancillary diagnostic tool” in conjunction with existing clinical routines.

However, we know that brain activity is constantly fluctuating and changing, even in the presence of repeating, external stimuli. Therefore, some variance between sessions should be expected in the fMRI signal, even in the presence of absolutely no noise.

In fact, we recently published a paper in which we make the very point of the reviewer.

https://journals.plos.org/plosone/article?id=10.1371/journal.pone.0299753

In Figure 1c of the publication, we demonstrate a correlation between the within-subject test-retest reliability of response behavior—interpreted as a measure of behavioral fluctuations—and the within-subject reliability of brain responses. This suggests the presence of an independent variable that causes fluctuations in both brain activity and response behavior that is not under the control of an indvidual. This observation holds clinical significance, as some individuals may be inherently unsuitable for the clinical applications of fMRI. Therefore, it is crucial to assess the test-retest reliability of patients undergoing pre surgical mapping.

If the gold standard time course is the predictor time course, shouldn’t this be plotted in Figures 6 and 7 so that we can visually evaluate which pipeline is the closest to what we would expect?

The primary objective was to demonstrate that the time courses that were filtered with the Savitzky-Golay (SG) method closely resemble the original data, whereas this is less true for time courses filtered with SPM low-pass filters. The comparison suggested by the reviewer is now illustrated in the new Figure 8, which presents a comparison of the power spectra of the predicted time course and the noise that remains after applying the SG and SPM filters.

Figure 6: I think the top figure plots the “residual” time courses, correct? If so, this should be mentioned since it is unclear as written.

The reviewer is correct; the original caption was indeed too brief. We have now revised it to provide a more detailed explanation. The blue signal represents the raw fMRI data, which displays a significant trend that must be addressed through filtering. In the updated caption, we compare how the Savitzky-Golay (SG) and SPM filters follow the trend present in the raw fMRI signal.

It seems like the individual HRF and “event-related average” are used somewhat interchangeably. This seems correct to me, but it would be clearer if one term was defined and used.

We now use the phrase event related average only twice to explain how the empirical HRF was obtained.

Reviewer #2: Reproducibility assessment of fMRI is a common practice at group level. Conventional signal post processing of fMRI has shown poor reproducibility on an individual level. The study aims to introduce a data driven signal filtering workflow with Savitzky-Golay filters to fMRI signal for improving single subject reproducibility in comparison SPM’s default signal filtering method. By improving subject level reproducibility, the work aims to improve the potential value of fMRI scans in clinical assessments.

Overall the experiments are documented in great detail and comprehensive. However, the current workflow is focused on SPM based workflow and described in some SPM specific terms. To allow the research to have a wider reach in the general fMRI research community, I have some suggestions to allow this work to be adopted by FSL, fMRIPrep, and AFNI users.

We strongly support initiatives like fMRIPrep, as they may lead to the standardization of analysis pipelines. However, as the reviewer points out, the current paper is a mix of the FSFAST pipeline available in FreeSurfer, some routin

---

## [Decision Letter · Decision Letter 1]

3 Mar 2025

Towards clinical applicability of fMRI via systematic filtering

PONE-D-24-38474R1

Dear Dr. Koten,

We’re pleased to inform you that your manuscript has been judged scientifically suitable for publication and will be formally accepted for publication once it meets all outstanding technical requirements.

Kind regards,

Kendrick Kay

Academic Editor

PLOS ONE

Additional Editor Comments (optional):

Dear authors,

We are in principle happy to accept your manuscript, provided you make the minor suggested revisions from R1.

Kendrick Kay

PLOS ONE Academic Editor

Reviewers' comments:

Reviewer's Responses to Questions

**Comments to the Author**

1. If the authors have adequately addressed your comments raised in a previous round of review and you feel that this manuscript is now acceptable for publication, you may indicate that here to bypass the “Comments to the Author” section, enter your conflict of interest statement in the “Confidential to Editor” section, and submit your "Accept" recommendation.

Reviewer #1: (No Response)

Reviewer #2: All comments have been addressed

2. Is the manuscript technically sound, and do the data support the conclusions?

Reviewer #1: Yes

Reviewer #2: Yes

3. Has the statistical analysis been performed appropriately and rigorously? 

Reviewer #1: Yes

Reviewer #2: Yes

4. Have the authors made all data underlying the findings in their manuscript fully available?

Reviewer #1: Yes

Reviewer #2: Yes

5. Is the manuscript presented in an intelligible fashion and written in standard English?

Reviewer #1: Yes

Reviewer #2: (No Response)

6. Review Comments to the Author

Reviewer #1: Overall, my comments have been largely addressed, and I only have a few minor comments left.

I would like to thank the authors for their detailed response to my first comment. I think that some brief text should be added to the based on this response, i.e. why the reliability of SPM pipeline with low-pass filters may be inflated or have other shortcomings.

I see that Figure S10 was added to address my concern on statistical comparisons. However, there is not sufficient information to interpret the figure. If the dot is warm-colored, does this mean that the first or second pipeline had higher reliability? Based on the result, I am assuming that warm colors indicate the first pipeline listed is higher and cool colors indicate the opposite. This information could just be added to the caption.

What are the units of the Figure 5-8 y-axis labels “Power” and “Bold Response”? Are they in percent signal change, Z-score, or some other normalized amplitude?

Reviewer #2: The authors has used PCA component, and PCA to refer to principal component (PC) in the method section. Please correct these!

I am happy with all the responses.

7. PLOS authors have the option to publish the peer review history of their article (what does this mean? ). If published, this will include your full peer review and any attached files.

**Do you want your identity to be public for this peer review?** For information about this choice, including consent withdrawal, please see our Privacy Policy .

Reviewer #1: No

Reviewer #2: No

---

## [Editor Report · Acceptance letter]

PONE-D-24-38474R1

PLOS ONE

Dear Dr. Koten,

I'm pleased to inform you that your manuscript has been deemed suitable for publication in PLOS ONE. Congratulations! Your manuscript is now being handed over to our production team.

Kind regards,

on behalf of

Dr. Kendrick Kay

Academic Editor

PLOS ONE